# Proline Isomerization: From the Chemistry and Biology to Therapeutic Opportunities

**DOI:** 10.3390/biology12071008

**Published:** 2023-07-14

**Authors:** Deepti Gurung, Jacob A Danielson, Afsara Tasnim, Jian-Ting Zhang, Yue Zou, Jing-Yuan Liu

**Affiliations:** 1Department of Medicine, University of Toledo College of Medicine, Toledo, OH 43614, USA; 2Department of Cell and Cancer Biology, University of Toledo College of Medicine, Toledo, OH 43614, USA; 3Department of Bioengineering, University of Toledo College of Engineering, Toledo, OH 43606, USA

**Keywords:** proline isomerization, post-translation modifications, proline isomerase, cyclophilin, cyclosporin, FK506 binding protein (FKBP), Interleukin-2 inducible T-cell kinase (Itk), 5-hydroxytryptamine type 3 (5-HT3), sanglifehrin, parvulin, Protein Interactor with NIMA1 (Pin1), Ataxia telangiectasia and Rad3-related (ATR), autoimmune disease, cancer, infectious disease, neurodegenerative disease, multiple sclerosis (MS), human immunodeficiency virus 1 (HIV-1), hepatitis C virus (HCV), Parkinson’s disease (PD), hepatitis B virus (HBV), systemic lupus erythematosus (SLE), Alzheimer’s disease (AD)

## Abstract

**Simple Summary:**

Proline isomerization influences protein folding and function tremendously and serves as a unique type of post-translational modification that regulates multiple biological pathways. Although impactful, the importance and prevalence of proline isomerization as a regulation mechanism in biological systems has not been fully understood or recognized. Aiming to fill gaps and bring new awareness, in this wholistic review, we attempt to connect various aspects of proline isomerization from its chemistry, historic discovery, and biological function, to related diseases. Therapeutic opportunities opened up by this unique behavior of proline and future urgent needs pertinent to the topic are clearly communicated.

**Abstract:**

Proline isomerization, the process of interconversion between the *cis*- and *trans*-forms of proline, is an important and unique post-translational modification that can affect protein folding and conformations, and ultimately regulate protein functions and biological pathways. Although impactful, the importance and prevalence of proline isomerization as a regulation mechanism in biological systems have not been fully understood or recognized. Aiming to fill gaps and bring new awareness, we attempt to provide a wholistic review on proline isomerization that firstly covers what proline isomerization is and the basic chemistry behind it. In this section, we vividly show that the cause of the unique ability of proline to adopt both *cis*- and *trans*-conformations in significant abundance is rooted from the steric hindrance of these two forms being similar, which is different from that in linear residues. We then discuss how proline isomerization was discovered historically followed by an introduction to all three types of proline isomerases and how proline isomerization plays a role in various cellular responses, such as cell cycle regulation, DNA damage repair, T-cell activation, and ion channel gating. We then explore various human diseases that have been linked to the dysregulation of proline isomerization. Finally, we wrap up with the current stage of various inhibitors developed to target proline isomerases as a strategy for therapeutic development.

## 1. Introduction

Isomers are a group of molecules that have the same composition of atoms but differ in the 3-dimensional arrangement of these atoms. Isomerization is a process in which one isomer undergoes structural changes to produce a different isomer. Isomerization can occur through bond rotations or rearrangement reactions. *Cis*/*trans* isomerization, also known as geometric isomerization, is a type of isomerization in which a molecule changes its spatial arrangement around a double bond or a ring structure. In a *cis*-isomer, the functional groups of interest are on the same side of a double bond or a ring, while in a *trans*-isomer, they are on opposite sides. *Cis*/*trans* isomerization switches a *cis*-isomer into a *trans*-isomer and vice versa.

Amino acids in a polypeptide chain can adopt either *cis*- or *trans*-conformation around the peptidyl partial double-bond. This can be denoted by an omega angle of 0° or 180°, respectively (Figure 1). The *cis*-conformation refers to a configuration where the two C^α^ atoms of the two connected adjacent amino acids are located on the same side of the peptidyl bond, whereas the *trans*-conformation refers to a configuration where they are on opposite sides of the bond. The two C^α^ atoms and their connected R groups are usually much bulkier than the hydrogen and oxygen atom around the partial peptidyl double bond and, as a consequence, cis-conformation is generally less energetically favorable due to steric hindrance leading to a strong *trans*-conformation preference by almost all other standard amino acids except for proline (Figure 2A,B), due to its special ring structure which is composed of a nitrogen atom bonded to the α- and delta-carbon (Figure 1B). Compared with the other 19 standard amino acids, additional phi angle constraints are exerted on proline due to this ring structure (Figure 1B), leading to a distinctive kink in the polypeptide chain that can be disruptive to regular α-helical or β-sheet secondary structures. Consequently, proline has a profound impact on protein folding, stability, and function. In addition, the unique ring structure also leads to another remarkable property of proline, that is its weaker preference to the *trans*-conformation around the peptidyl bond connecting it to its preceding amino acid residue. This is because the cyclic structure of proline loops the side chain atoms back to the backbone and brings either its C^α^ or C^δ^ to the same side of the C^α^ atom of the preceding residue in *cis*- or *trans*-conformation, respectively (Figure 2C,D), leading to a much smaller difference in steric crowdedness between these two conformations compared with the difference seen in linear residues (Figure 2A,B). As a result, proline residues can adopt a significant abundance of both *cis*- and *trans*-conformations. 

## 2. The Discovery and Study of Proline Isomerization in History: From Initial Observations to Modern Insights

The study of proline has its roots in a landmark discovery by Richard Willstatter in 1900 [1]. He synthesized proline from sodium malonic ester and trimethylene bromide. A year later in 1901, Emil Fischer isolated proline from hydrolyzed casein [2]. Since then, the characterization of the peptide bond and depiction of atomic structures of amino acids including proline have revealed the existence of *cis*- and *trans*-conformers. It was not until several decades later, in 1975, that Brandts and his colleagues showed the role of proline isomerization as a rate-limiting step in the unfolding and refolding of bovine ribonuclease A [3]. In their research, they conducted double-jump experiments to investigate the behavior of the enzyme. Initially, the enzyme solution in its native state was exposed to a denaturing condition by lowering the pH to 2.0 at 51 °C. In a subsequent experiment, the enzyme was exposed to a refolding condition by raising the pH to 4.5 at 0 °C. By being able to observe kinetic phases during the transition period, they found that proline isomerization was a major contributing factor to the gradual step in the denaturation reaction. This was probably the first work showing the biological relevance of proline *cis*/*trans* isomerization. Additional in-depth investigations have further supported the significant impact of proline isomerization on the kinetics of protein folding over the years [4,5,6].

In the mid-1900s, X-ray crystallography was used to determine the structure of myoglobin [7] and lysozyme [8] and then became an important tool to study the structures of proteins in the following years. X-ray crystallography, being a static snapshot of the protein crystal structure, was not initially able to capture the dynamic process of isomerization or conformational variations in proteins. However, as a greater quantity of crystal structures became available, it was clear that one protein can adopt multiple conformations. Smith et al. initially examined the crystal structures of the proteins crambin, erabutoxin, myohemerythrin, and lamprey hemoglobin deposited in the Protein Data Bank (PDB) and discussed their intrinsic conformational variability [9]. They suggested that structures determined at a high resolution (≤2Å) more readily detect structural variations. Multiple and distinct conformations for a few amino acid side chains including proline were revealed in protein crystals in their native state. This significant observation paved the way for researchers to understand protein flexibility at the level of amino acid side chains. Breakthrough work on characterizing *cis*/*trans* peptide bonds in protein structures was carried out by Stewart et al. in 1990 [10]. They studied about 150 protein crystalline models and about 33,000 peptide bonds from the PDB and observed an increased prevalence of *cis*-peptide bonds, particularly at Xaa-Pro located in bends and turns. This implies proline may potentially play a role in structural heterogeneity. In addition, the group also postulated a possible bias toward the *trans*-conformation in the molecular-structure-modeling step of X-ray crystallography. In 1999, Pal and Chakrabarti compared *cis*-peptide bonds containing proline and non-proline residues in 147 protein crystal structures and showed the prevalence of *cis*-proline nearby aromatic residues due to their stabilization effect via CH---pi interactions [11]. In 2002, Reimer and Fisher reported that two distinct isomers were identified in nearly 4% of the ~800 non-homologous protein crystal structures due to proline *cis*/*trans* isomerization [12]. They showed local structural rearrangement extending through the protein backbone from proline *cis*/*trans* isomerization site to its neighboring region. Interestingly, they also found that protein-protein interactions involving proline-containing motifs are conformer-specific, rendering specific biological functionality to one of the conformers. Finally, Pahlke et al. found that *cis*-prolines occur more frequently in surface-accessible areas [13].

Compared with X-ray crystallography, nuclear magnetic resonance (NMR) spectroscopy has proven to be more successful in the study of proline isomerization, although it works only for small-sized proteins. The sample being in the solution state increases its physiological relevance and facilitates the unbiased observation of the *cis*/*trans* populations. Additionally, NMR is a rate-sensitive method as the rate of peptidyl-prolyl *cis*/*trans* isomerization is slower than the NMR chemical shift integration time scale, thus, the shift of the two resonance frequencies of isomers can be seen in NMR spectra when triggering factors are introduced [14]. In 1981, Grathwohl and Wuthrich used ^13^C NMR to study the *cis*/*trans* conformers in simple polypeptides containing only alanine and proline in different lengths [15]. They showed that ^13^C NMR can experimentally determine the ratio of *trans/cis*-isomers. Using their method, they observed that this ratio can change when the pH of the solution changes for some peptides. Their experiments also confirmed that the dominant form of either *cis*- or *trans*-isomer is sensitive to the nature of the preceding residues, namely, the type of Xaa in peptides containing Xaa-Pro. In the following decade, consecutive studies using ^1^H and ^13^C NMR spectroscopy on collagen [16], staphylococcal nuclease [17], insulin [18], and calbindin D9k [19,20] revealed the existence of multiple conformations of these proteins due to *cis*/*trans* isomerization. Advancements in 2D and 3D NMR techniques have led to a deeper understanding of the role *cis*- and *trans*-proline populations play in structural changes of proteins. Recently, Sebak et al. devised a ^1^H NMR technique that boasts enhanced sensitivity, enabling the detection of even the rarest conformers in 2022 [21]. To this day, NMR remains the most effective and widely used direct approach to detect *cis*/*trans* isomerization in proteins. In addition, researchers have combined NMR and X-ray crystallography to validate the findings from each individual method. For example, in the study of staphylococcal nuclease [22], and calbindin D9k [23], high resolution crystal structures were found to be agreeable with their previously reported NMR data [16,18,19,20,24].

In addition to NMR and X-ray crystallography, which provide direct and hard evidence, there are several other indirect methods that can be used to detect possible *cis*/*trans* isomerization in proteins. For example, proline isomerization events that cause *cis*/*trans* population shift can be inferred by their biological consequences. These can be detected by molecular biology methods such as Western blot, pull-down assay, co-immunoprecipitation, etc. Recently, surface plasmon resonance combined with molecular dynamics simulations [25] and mass spectrometry [26,27,28] have been shown to hold promising results. Furthermore, although challenges remain, multiple attempts using bioinformatics and machine learning algorithms to predict *cis*/*trans* isomerization in proteins have been made [29,30,31,32,33]. 

Today, proline isomerization is recognized as a crucial mechanism in regulating cellular processes and is an area of active research within the fields of biochemistry and molecular biology. Despite many advances, there is still much to be learned about proline isomerization, making it an exciting and rapidly evolving area of scientific inquiry. The continued exploration of proline isomerization holds the promise of unlocking new insights into the fundamental processes that govern life.

## 3. Proline Isomerization Catalyzers: Peptidyl-Prolyl Isomerases 

Proline isomerization is a constant interconversion process that goes from *cis* to *trans* and vice versa. Over time, the concentrations of *cis*- and *trans*-conformers become constant and the rates of proline isomerization in both directions become equal and the system reaches dynamic equilibrium. The ratio of the concentrations of the *trans*- to *cis*-isomer at equilibrium is called the equilibrium constant (K_eq_) and can be given by the following equation: K_eq_ = [*trans*]/[*cis*]. The value of K_eq_ quantifies the relative preference of either conformer under a given condition. K_eq_ > 1 indicates the *trans*-conformer is preferred, while K_eq_ < 1 indicates the *cis*-conformer is preferred. A K_eq_ = 1 indicates both conformers are equally favorable. At a given condition at equilibrium, K_eq_ is a constant. However, K_eq_ is a condition-dependent parameter and varies when the environment or the protein structure are changed. These changes upset the equilibrium leading to unequal interconversion rates between *cis*- and *trans*-conformers until a new equilibrium constant, K_eq_’, is reached. Due to the small energy difference between the proline *cis*- and *trans*-conformers, it is common to see that at one condition, the *trans*-conformer is dominant (K_eq_ > 1) while at a different condition, *cis* is more favorable (K_eq_’ < 1). When *cis*/*trans* conformational changes are mentioned in this paper, this refers to an inversion of the dominant conformer in a new equilibrium state. 

Proline isomerization is a very slow interconversion process at a scale of 2.5 × 10^−3^ s^−1^ at 25 °C [15]. This rate-limiting step prevents a quick shift from *cis* to *trans* or vice versa when a triggering factor is present. However, this process can be accelerated by a group of enzymes called peptidyl-proline isomerases (PPIases), which catalyze the conversion by a factor of 10^3^–10^6^ fold [34]. It is important to note that PPIases do not change the equilibrium constant at a given condition but accelerate the speed of reaching a new equilibrium state when a triggering factor disrupts the original equilibrium. Based on substrate specificity, PPIases have been categorized into three groups: cyclophilins, FKBPs and parvulins. Each type of PPIase shows sequence-specificity for Xaa-Pro motifs and catalyzes proline isomerization in its protein substrates; however, there is no sequence homology among these types of PPIases. Furthermore, the active sites of these PPIases, and the structure of the substrates/ligands they bind to, are also different in structure. All PPIases act like a switch by accelerating the conversion to the dominant isomer form of proline when conditions are changed, and are involved in the regulation of protein folding, activation, and/or degradation [14]. Below, we briefly introduce each group of PPIases.

### 3.1. Cyclophilins 

Cyclophilin A (CypA) was the first PPIase discovered during the search for the protein target of the immunosuppressive drug cyclosporin A (CsA), in 1984 [35]. Its peptidyl-prolyl catalytic activity was described in the same year and was shown to accelerate protein refolding [35,36] and the crystal structure of CypA bound with CsA was solved after about a decade [37]. Cyclophilins are members of the immunophilin family, and they have been highly conserved throughout their evolution from bacteria to human [38]. The human genome encodes 18 different cyclophilin genes, and 7 of them (CypA, CypB, CypC, CypD, CypE, Cyp40, CypNK) have been well characterized [39]. Each cyclophilin consists of at least one cyclosporin-binding domain (CLD) that is 109 amino acids long and contains the PPIase activity.

CypA is one of the most dominant cyclophilin isoforms in human tissues and is abundant in the cytosolic fraction of T-cells [40]. It is also the smallest cyclophilin that contains a single CLD. The three-dimensional structure of CypA reveals that it folds into an eight-stranded anti-parallel β barrel with one α-helix on both sides (Figure 3A). It has been demonstrated that critical residues in the active site include Phe60 which correctly positions the prolyl ring of the substrate and Arg55 that may stabilize the sp^3^ hybridization of the proline nitrogen atom during the transition state and thereby facilitate the turning of the omega angle [41]. In addition to CLD, some cyclophilins have unique domains that determine their sub-cellular localization and functions. For instance, CypD is localized in mitochondria because of its mitochondrial localization signal domain in the N-terminal [42]. Larger cyclophilins, such as Cyp40, have a C-terminal tetratricopeptide repeats (TPR) domain that binds to heat-shock proteins and other protein chaperones in response to cellular stress [43]. Cyclophilins are ubiquitously expressed in human tissues and are involved in the regulation of protein folding, protein complex assembly [44] and trafficking [45], signal transduction [46,47,48], T-cell activation [49,50], and mitochondrial permeability transition [51]. Detailed information about the structure and function of cyclophilins can be found in other reviews, such as those by Galat 2004 [39], Wang et al. 2005 [38], and Davis et al. 2010 [52].

### 3.2. FK506 Binding Proteins (FKBPs) 

During the search to identify the protein target and elucidate the mode of action of the immunosuppressive drug FK506 (tacrolimus), researchers discovered a 12-kDa protein in T-lymphocytes. This protein binds to FK506 and was subsequently named FK506 binding protein (FKBP) [57,58]. In addition to FK506, this FKBP, later referred to as FKBP12, was also found to bind to another immunosuppressive drug, rapamycin (sirolimus) [59]. FKBP12-FK506 and FKBP12-rapamycin complex binds to calcineurin and mammalian target of rapamycin (mTOR), respectively [59,60]. Wilson et al. determined the crystal structure of FKBP12 in complexed with FK506 and rapamycin and confirmed the earlier observations in 1995 [54]. The three-dimensional structure of FKBP12 reveals an overall structure of five-stranded antiparallel β-sheets with a short α-helix (Figure 3B).

Like cyclophilins, FKBPs are a group of highly conserved proteins belonging to the immunophilin family. They are present in all eukaryotes from yeast to human. In human, there are several FKBP homologs, but only a few of them (FKBP12, FKBP13, FKBP25, FKBP51, FKBP52) have been well characterized to date [61]. They are named based on their molecular weight and are subdivided into four groups; cytoplasmic (FKBP12, FKBP12.6, FKBP51, FKBP52), endoplasmic reticulum (FKBP13, FKBP19, FKBP22, FKBP63, FKBP65), nuclear (FKBP25, AIPL1) and TPR-containing FKBPs (FKBP36, FKBP37, FKBP38, FKBP44, FKBP51, FKBP52) [39,61,62]. One of the differences between FKBP12 and larger FKBPs such as FKBP51 and FKBP52 is that the larger FKBPs consist of two or more consecutive FK506 binding domains. All FKBPs possess at least one FK506 binding domain in their N-terminal region, which serves for both PPIase activity and FK506 binding. The PPIase domains of most FKBPs exhibit high sequence homology and similarities in their tertiary structures. Apart from the PPIase domain, each FKBP contains unique domains that determine their specific function and subcellular localization such as the TPR domain (FKBP51, FKBP52) which mediates protein-protein interactions and the calmodulin-binding domain or EF-hand motif (FKBP22, FKBP23, FKBP65, FKBP66) that binds to multiple calcium ions. In mammals, FKBPs are expressed in most tissues and therefore play diverse roles in several biochemical processes including protein folding and trafficking, receptor signaling, transcription control, apoptosis, and T-cell activation [63,64]. More detailed information regarding the structure and function of each FKBP can be found in the reviews by Tong and Jiang [65] and Kolos et al. [61].

### 3.3. Parvulins

The first parvulin was discovered in *E. coli*, and is only 92 amino acids long, which makes it the smallest PPIase known so far [66,67]. Its PPIase activity was shown by monitoring the time course of the difference in absorbance at 390 nm and 520 nm using a protease-coupled enzymatic assay. The tetrapeptide succinyl-Ala-Leu-Pro-Phe-4-nitroanilinde was used as the substrate which releases 4-nitroanilinde from its C-terminal by α-chymotrypsin digestion only if the tetrapeptide is in the *trans*-conformation. The appearance of 4-nitroanilinde was detected by its absorbance at 390 nm. The absorbance at 520 nm was also monitored as a reference because the absorbance at this wavelength is constant. α-chymotrypsin’s high specificity for the *trans*-substrate requires that any increase in the kinetic rate is directly proportional to the conversion from *cis*- to *trans*-conformation. It was determined that the tetrapeptide substrate exists at a *cis:trans* ratio of around 1:5 at pH 7.8. In 1997, Scholz et al. showed that parvulin is involved in the folding of ribonuclease T_1_ protein of *Aspergillus oryzae* by catalyzing proline isomerization on Pro39 and Pro55 to their *cis*- form from the unfolded *trans*-intermediate of the protein.

The ortholog of parvulin in humans is called Pin1 (Protein Interactor with NIMA1) or Par18, which was discovered by Lu et al. [68] while investigating the regulatory mechanisms of cell division and cell cycle in 1996. They employed a two-hybrid yeast assay to show that Pin1 interacts with NIMA (Never in Mitosis A) and is involved in cell cycle control. Other than Pin1, Par14 and Par17 are two other members in this family. Par14 and Par17 are different splicing isoforms both encoded by Pin4 gene. Par14 is a 13 kDa protein discovered by Uchida et al. and Rulten et al. in 1999 [69,70]. Par17 differs from Par14 by the presence of an extra 25 residues at the N-terminal [71,72]. Unlike FKBPs and cyclophilins, parvulins are not associated with immunosuppression, but are also highly conserved from prokaryotes to human along with their conserved PPIase domain, which is similar to the other two types of PPIases. 

Among all parvulins in human, Pin1 is the most well-studied. Pin1 binds to a wide range of substrates, influencing protein folding, stability, and subcellular localization, and is involved in a variety of cellular processes such as cell cycle progression, transcription response to DNA damage and stress, and immune responses by regulating multiple cellular signaling pathways. It consists of two domains: a WW domain in the N-terminal region that binds to phosphorylated serine or threonine followed by a proline motif, and a PPIase domain in the C-terminal region that possesses peptidyl-prolyl isomerase activity (Figure 3C). The crystal structure of the PPIase domain of Pin1 is composed of four antiparallel strands of β-sheets surrounded by α-helices [55]. Pin1 is unique in its ability to specifically bind to and isomerize phosphoproteins at the pSer/pThr-Pro site. The substrate-binding pocket of Pin1 is therefore unique due to the presence of highly conserved basic residues (Lys63, Arg68, and Arg69) that enable the PPIase domain to recognize substrates in a phosphorylation-dependent manner [73,74]. 

Par14 and Par17 are distinct from Pin1 in that the WW domain is replaced by an N-terminus DNA-binding domain [75]. Additionally, in contrast with Pin1, Par14 and Par17 prefer arginine or leucine residues preceding proline [76]. Burgardt et al. have shown that Par17 is involved in calcium ion and calmodulin-dependent tubulin polymerization in 2015 [77]. Very recently, Kim showed that Par14/17 catalyzes isomerization of the Arg133-Pro134 motif of Hepatitis B core protein and upregulates hepatitis B virus (HBV) replication in 2023 [78]. The cellular function of Par14 and Par17 is still being explored. Further details regarding these parvulins can be found in other reviews, such as those by Lu et al. [79], El Boustrani et al. [80], Matena et al. [72] and Chen et al. [81]. A comparison of all three families of PPIases is summarized in Table 1.

## 4. Tying It Altogether: Showcase of Proline Isomerization as a Regulatory Mechanism in Cellular Response and Function

Many studies have shown that proline isomerization is involved in regulating signal transduction and thus cellular functions. Proline isomerization is widely accepted as a non-covalent modification of peptide backbone that is different from covalent post-translational modifications that happen on the side chains of a specific amino acid residue such as phosphorylation or acylation. Proline isomerization is unique in that the structural changes happen on the main chain, and it involves multiple players beside the protein being switched between *cis*- and *trans*-conformation around a specific proline. Collectively, these factors impact cellular functions via regulation of cellular processes such as DNA repair, cell cycle, apoptosis, and immune response. Below, some proteins are showcased to exhibit proline isomerization being utilized as a regulatory mechanism in cellular response and function.

### 4.1. ATR Is Regulated by Proline Isomerization to Switch between Its Dual Role in Modulating Cell Death and DNA Damage Checkpoint Signaling 

Ataxia telangiectasia and Rad3-related protein (ATR) is a member of the serine/threonine phosphoinositide-kinase-related-3 kinases that phosphorylates various substrate proteins, including checkpoint kinase 1 (Chk1) and p53, in response to DNA damage agents. ATR plays a crucial role in DNA damage response (DDR) as a master regulator, and it is involved in checkpoint signaling pathways to maintain the stability and integrity of genomes [82]. Structurally, ATR consists of several domains, including the N-terminal domain Huntington, Elongation Factor 3, PR65/A, TOR (HEAT) repeats (N-HEAT) and middle HEAT (M-HEAT), BH3 domain, the FRAP-ATM-TRRAP (FAT) domain, the kinase domain, and the C-terminal domain with FATC, with the latter three domains bearing catalytic activities. 

ATR has been reported to have two distinct conformations due to proline isomerization at the Ser428-Pro429 motif [83,84,85]. The roles of these two conformations are different: *cis*-ATR is located in mitochondria at outer membrane of the cytoplasm and is anti-apoptotic and probably supports oncogenesis in certain circumstances, whereas *trans*-ATR is located in the nucleus and is the critical kinase involved in modulating DNA damage checkpoint signaling and maintaining genomic integrity (Figure 4A). ATR is naturally more stable in the *cis*-form in the cytoplasm, and it carries an exposed/activated BH3 domain that can bind to tBid and prevent the activation of pro-apoptotic molecules such as Bax or Bak, and therefore *cis*-ATR is anti-apoptotic. The shift from *cis*- to *trans*-ATR is triggered by the phosphorylation on Ser428. Pin1 recognizes the pSer428-Pro429 motif and accelerates the conversion to *trans*-ATR [86]. Once being converted, *trans*-ATR is transported to the nucleus and binds to ATRIP. In response to DNA damage, ATR auto-phosphorylates on its own Thr1989 in the FAT domain in the nucleus and recruits TopBP1 to become further activated, triggering a cascade of events involving various critical downstream proteins such as p53 and checkpoint kinases such as Chk1. This eventually leads to cell cycle arrest in the S-phase, resulting in either DNA damage repair or apoptosis depending on the severity of the DNA damage [87,88]. The cryo-EM structure of the ATR-ATRIP complex reveals that ATRIP interacts with ATR monomers through the N-HEAT on the N-terminus [86]. It has been shown that upon DNA damage, such as UV, protein phosphatase 2A (PP2A) can dephosphorylate the Ser428 site leading to an increased level of cytoplasmic *cis*-ATR and subsequently enhance anti-apoptotic activities. The distinct functions of *cis*- and *trans*-ATR require a delicate balance of the levels of these two forms to ensure cellular homeostasis that prevents *cis*-ATR being exploited by cancer cells for their survival while maintaining genome integrity and stability in normal cells. 

### 4.2. p53 Is Regulated by Proline Isomerization at Multiple Sites in Response to DNA Repair and Cellular Stress, Respectively

p53 is a transcription factor and a tumor suppressor that controls cell fate by regulating critical cellular processes such as DNA repair, cell cycle, and apoptosis through various downstream signaling pathways. p53 exists as a homo-tetramer with each subunit consisting of a transactivation domain (TAD 1/2) in the N-terminal region, followed by a proline-rich region (PRR) located between positions 61 and 94, a DNA binding core domain, a tetramerization domain in the middle, and a carboxy terminus domain (CTD) in the C-terminal region [90]. The protein is tightly regulated in healthy cells, with proline isomerization identified as an important mechanism that regulates the protein in multiple folds.

Firstly, proline isomerization is a regulation mechanism for p53 stability. In response to DNA damage, p53 undergoes phosphorylation at its different serine/threonine sites, resulting in increased amount and transcriptional activity that leads to cell cycle arrest. Zheng et al. first described how p53 activation depends on Pin1 in response to genotoxic insult [91]. They showed that DNA-damage-mediated phosphorylation of Ser/Thr-Pro motifs promotes the interaction of Pin1 with p53. Numerous studies have since sought to clarify how Pin1 regulates p53 with the two main effects of Pin1 on phosphorylated p53 noted: (i) Pin1 increases the half-life of p53 and (ii) Pin1 enhances DNA binding activity and transcriptional activity of p53 with its target genes (such as p21, Bax). More recently, the exact molecular basis of regulation of p53 via proline isomerization was elucidated. Under normal and unstressed conditions, the p53 level is at minimum. Under DNA damage, Pin1 catalyzes the *cis*/*trans* isomerization at several phosphorylated Ser/Thr-Pro sites (Ser33, Ser46, Thr81 and Ser315) on p53 resulting in a weaker binding affinity to Mouse double minute 2 homolog (MDM2), a E3 ubiquitin ligase which signals to degrade p53, and thereby rescues p53 from degradation [92,93]. The theoretical difference of affinity between the *cis*- and *trans*-p53 with MDM2 has been estimated to be around 2 kcal/mol [94]. Additionally, Zheng et al. showed that phosphorylation on p53 at all three sites, pSer33, pThr81 and pSer315, is required for Pin1 to bind and to enhance the transcriptional activity of phosphorylated p53 via increasing its affinity to the DNA element, in particular, the p21 promoter region, resulting in p53-induced G1 cell cycle arrest in response to DNA-damage [91]. This observation is in agreement with the discovery of another group who observed that Pin1 binds to phosphorylated p53 at pSer33 and pSer46 in response to radiation DNA damage in mouse embryo fibroblasts, and cells co-transfected with Pin1 and p53 showed enhanced p21 transactivation compared to cells transfected with Pin1 alone [92]. 

In addition, proline isomerization has also been identified as a mechanism for p53 to exert its regulatory effects on apoptosis under cell stress via direct interaction with Bax, which is an apoptotic effector known to be traditionally activated by BH3 domain-containing proteins such as BIM and BID. However, p53 activates Bax by a different mechanism [95]. Under cell stress, Ser46 of p53 is phosphorylated, and Pin1 subsequently recognizes the pSer46-Pro47 motif and catalyzes *cis*- to *trans*-proline isomerization at the Pro47 site of p53. The conformational change in p53 results in the release of α helices 6 through 9 of Bax, a characteristic structural displacement required for Bax activation [95]. 

Finally, apart from Pin1, research has revealed that certain cyclophilins, namely CypD and Cyp18, are involved in phosphorylation-independent proline isomerization of p53 in response to DNA damage or cellular stress. Specifically, molecular dynamics simulations predicted that CypD binds to p53 at Pro151 in the DNA binding domain [96], while another study suggests that it binds to the N-terminal domain (NTD) of p53 [97]. The interaction between CypD and p53 is believed to facilitate p53-mediated opening of mitochondria permeability transition pores and programmed necrosis, particularly in response to oxidative stress [98], although the detailed mechanism is still elusive. In addition to CypD, it has been reported that Cyp18 may bind to Pro72 in p53, reducing its DNA-binding activity and inhibits its apoptotic activity in T-cells [99]. The contrasting effects of Pin1 and Cyp18 on p53 can be attributed to the fact that DNA damage was not induced in the latter case. This suggests that the influence of Pin1 and Cyp18 on p53 may be context-dependent and could vary depending on the presence or absence of DNA damage.

### 4.3. Itk Is Activated by Proline Isomerization in T-Cells

Interleukin-2 inducible T-cell kinase (Itk) is a non-receptor (Tec family) tyrosine kinase that plays a critical role in the development and differentiation of T-cells. Itk is highly expressed in immune cells including T-cells, mast cells, and Natural Killer cells. Itk is a cytosolic enzyme composed of five distinct domains: the N-terminal pleckstrin homology (PH), Tec homology (TH), Src homology 3 (SH3), Src homology 2 (SH2) and the C-terminal kinase catalytic domain. Itk has been recognized as a positive mediator in T-cell receptor (TCR) signaling [14]. Itk is activated upon T-cell receptor crosslinking, which occurs when the TCR binds to its specific antigen on the surface of an antigen-presenting cell or an infected cell. This interaction triggers a signaling cascade that leads to the activation of Itk, as well as other proteins involved in T-cell activation. After being activated, Itk triggers the phosphorylation and activation of PLCg1, which then regulates TCR stimulation by activating the Mitogen-activated Protein Kinase (MAPK) pathway and/or the Protein Kinase C (PKC) pathway, resulting in the production of interleukin 2 (IL-2) [100]. Research has indicated that dysregulation of Itk can lead to various T-cell/B-cell lymphoma, leukemia, and autoimmune diseases [14].

The regulation of Itk catalytic activity in T-cells is critical for proper T-cell signaling, and the SH2 domain of Itk plays a significant role in this process. Inactive Itk is self-inhibitory by forming a closed dimeric structure via self-association between the SH2 domain of one subunit and the SH3 domain of the opposing subunit [101]. When TCR is activated, signaling proteins such as SLP-76 are phosphorylated on certain tyrosine residues and the phosphotyrosines can compete to bind with the SH2 domain of Itk. This leads to the disruption of Itk homodimerization and activation of the Itk catalytic domain [102]. CypA is involved in the above regulatory process and the affinity of the SH2 domain to phosphorylated proteins or the SH3 domain has been shown to be greatly affected by proline isomerization. CypA can also interact with Itk through its SH2 domain and inhibits its kinase activity in the absence of TCR signals [101]. Under the CypA-bound state, Pro287 in the SH2 domain of Itk is in *cis*-conformation as suggested by NMR studies and the *cis*-SH2 domain favors homodimerization with the SH3 domain from the other Itk subunit. Upon TCR stimulation, CypA catalyzes the proline isomerization from *cis* to *trans* at the Asn286-Pro287 site within the SH2 domain (Figure 4B). In contrast with the *cis*-SH2 domain, the *trans*-SH2 domain preferentially binds to phosphotyrosine proteins [103]. As a result, the *cis*- to *trans*-isomerization on Itk in the SH2 domain by CypA disrupts Itk homodimerization and promotes the association of Itk to phosphotyrosine proteins [89,101,104]. Furthermore, binding of the phosphoprotein results in the release of CypA, causing structural perturbations in the catalytic domain of Itk, which activates its kinase activity and initiates the signaling cascade [101,105].

### 4.4. The Gate Function of 5-HT3 Receptor Is Regulated by Proline Isomerization

5-hydroxytryptamine type 3 (5-HT3) receptor is a membrane protein that belongs to the superfamily called Cys-loop ligand-gated ion channel and is activated by binding of 5-hydroxytryptamine, more commonly known as serotonin. Cys-loop ion channels consist of a characteristic loop formed by a varying number and sequence identity of amino acids bridged between two cysteine residues (Cys-Xaa_n_-Cys) near the N-terminal domain. 5-HT3 receptor is cation-selective and selectively permeable to sodium, potassium, and calcium ions [106]. The receptor consists of five subunits that are arranged around a central pore, which transports cations. Each subunit has an extracellular N-terminal domain that contains the neurotransmitter binding site, a transmembrane domain made up of four α-helices (M1, M2, M3, and M4) that form the ion channel pore, and an intracellular domain. The receptor opens the ion pore when an agonist, such as 5-hydroxytryptamine, binds to the extracellular domain and converts this chemical signal into an electric signal throughout the nervous system [107].

In an earlier investigation in 2001, Deane and Lummis identified several conserved proline residues within the 5-HT3 receptor with a small proportion of these prolines adopting a cis-conformation [108]. A different study by Lummis et al. in 2005 revealed that mouse 5-HT3 channels open upon ligand binding and are regulated by the *cis*/*trans* isomerization of a conserved proline residue, Pro308 (or Pro8* as noted in the paper) [109]. This proline residue is located at the interface between the extracellular and transmembrane domain (M2-M3 loop), which is believed to have a significant impact on the structure of the protein and its ability to gate the channel. The function of the receptor was found to be impaired when amino acids other than proline were introduced at this position through site-directed mutagenesis. In an additional mutagenesis study, Pro308 was substituted by cyclic proline analogues that bear varying degrees of preference for maintaining *cis*- or *trans*-conformations. The mutations with proline analogues that strongly favored the *trans*-conformation resulted in non-functional channels. A relationship between the free energy calculation difference of *cis*- and *trans*-isomers of the proline analogues and their activity of functional 5-HT3 receptors, as determined by the EC_50_ value for receptor activation, showed that those proline analogues which favored *cis*-conformation resulted in increased speed of 5-HT3 activation [109]. The role of Pro308 in the opening and closing of the channel was reinforced by ^1^H NMR experiments using peptides containing an M2-M3 loop [109]. In the closed state of the ion channel, Pro308 is in the *trans*-conformation. Upon binding of the ligand, Pro308 undergoes isomerization from *trans*- to *cis*-conformation, resulting in a structural change that reorients the M2 transmembrane helix and opens the channel. A molecular dynamics simulation study by Crnjar et al. in 2019 showed that isomerization of Pro 308 affects not only the structure of the receptor but also its electrostatic properties which decrease the electro potential of Asp271 that pulls Na^+^ ions towards the channel [110]. Even with this evidence, the role of proline isomerization in channel gating of the human 5-HT3 receptor homologue is still controversial. Paulsen et al. performed a mutagenesis electrophysiology study in 2009 on the human 5-HT3 receptor expressed in *Xenopus* oocytes [111]. This study showed that substitution to His or Trp of Pro303, which is the equivalent proline of Pro308 in the mouse homologue, did not substantially impair the channel gating process induced by serotonin, suggesting that *cis*/*trans* isomerization of this residue does not regulate channel gating. Pro303His and Pro303Trp resulted in the receptor currents being more sensitive to changes in the concentration of external Ca^2+^ compared to the wild-type receptor. Additionally, the mutations also caused the currents induced by 5-HT3 to desensitize more rapidly than those in the wild-type receptor, suggesting that Pro303 participates in controlling the sensitivity of the 5-HT3 receptor to its agonist through the rate of receptor desensitization [111]. However, it has also been suggested that the rates of activation of the 5-HT3 receptor are too fast to be driven by the conformational change of Pro303 and isomerization may not be the primary mechanism for channel gating [112]. This discrepancy can be solved if the isomerization process is catalyzed by some unidentified PPIases, which may sufficiently increase conformational change rate to the point required for ion gating regulation. Therefore, more studies are urgently needed to elucidate this mechanism to reach a definitive conclusion.

## 5. Role of Proline Isomerization in Human Disease

We have shown that proline isomerization can have a significant impact on protein structure and function and is employed as a mechanism in multiple biochemical processes and cellular responses. Therefore, the expression, enzymatic activity, post-translational modification, and localization of all PPIases are strictly regulated under normal physiological conditions. Malfunction or dysregulation of these enzymes can lead to various diseases. In this section, we discuss how proline isomerization and PPIases are linked to autoimmune diseases, cancers, viral infections, and neurodegenerative diseases. 

### 5.1. Autoimmune Disease

The recognition of the link between proline isomerization and immunity has a long-standing history as evidenced by the fact that the first PPIase, Cyclophilin A, was discovered during the search for the protein target of the immunosuppressive drug cyclosporin. It is therefore no surprise that research groups discovered that proline isomerization plays critical roles in modulating immune cell activation and immune tolerance, concluding that dysregulation of PPIase activity may contribute to autoimmunity. In this section, we focus on how PPIases are involved in the pathogenesis of some autoimmune diseases, such as multiple sclerosis, systemic lupus erythematosus, and rheumatoid arthritis. 

Multiple sclerosis (MS) is a chronic and progressive autoimmune disorder characterized by the immune system targeting the myelin sheath in the central nervous system. The hallmark features of MS include axonal injury, neuronal loss, and central nervous system atrophy [113,114], which are thought to result from the accumulation of reactive oxygen species, neuronal Ca^2+^ overload, and the activation of Ca^2+^-dependent cysteine proteases called calpains [115,116]. CypD is a key regulator of the mitochondrial permeability transition pore (mPTP), which is a non-specific channel in mitochondrial inner membrane that facilitates the entry of small molecules (<1.5 kDa) into mitochondria. The binding of CypD to mPTP leads to the formation of a channel that allows the influx of solutes up to 1.5 kDa, which can trigger mitochondrial swelling and rupture, and subsequent cytochrome C release and cell death (Figure 5A). In studies using an animal model for MS, CypD knockout mice showed better recovery from MS compared to control mice [117]. Additionally, selective inhibition of CypD in neuronal cells by a CypD inhibitor, JW47, showed significant protection of axons in the experimental model. This neuroprotective effect of JW47 was due to its direct inhibition of Ca^2+^-mediated mPTP formation [118]. Therefore, inhibition of CypD–mPTP interaction can protect against axonal injury and neurodegeneration, highlighting the potential of this pathway as a target for MS treatment. 

Recent research by Ge et al. in 2021 has shown that inhibition of Pin1 by juglone, a Pin1 inhibitor, can also significantly reduce inflammation and demyelination in a mouse model of MS called experimental autoimmune encephalomyelitis [119]. This effect was achieved through the suppression of CD4+ T-cells that produce IFN-β and IL-2, specifically Th1 and Th17 cells. It is well-established that CD4+ T-cells primed against myelin oligodendrocyte glycoprotein antigen are involved in MS pathogenesis [120]. Furthermore, Juglone treatment also resulted in the suppression of the costimulatory molecule CD38 in dendritic cells, suggesting a pathogenic role for Pin1 in MS [119]. In addition, myelin basic protein (MBP), a critical component of the myelin sheath in the central nervous system, contains proline-rich motifs and has been identified as a ligand for Fyn-SH3 (Src homology 3) domain [121]. This has led to the speculation that proline isomerization, specifically through Pin1, could potentially modulate MBP function and contribute to MS [122,123,124]. Evidently, future studies to elucidate the detailed mechanism and to explore Pin1 inhibition as a potential therapeutic strategy are essential to develop novel methods for MS treatment.

Systemic lupus erythematosus (SLE) is a systemic autoimmune disease that can affect multiple organs such as skin, lungs, heart, kidneys, muscles, and joints. One of the prominent molecular hallmarks in SLE patients is increased Type 1 IFN in their serum, which is produced primarily by myeloid cells [124,125]. The elevated production of Type 1 IFN leads to hypergammaglobulinemia in the following steps: (i) breakdown of immune tolerance, which normally prevents the immune system from attacking self-tissues, (ii) activation of autoreactive B cells, leading to the production of autoantibodies against self-antigens, (iii) autoantibodies then form immune complexes, which deposit in various tissues and organs, causing inflammation and tissue damage. In addition, type 1 IFN can also enhance the survival and activation of B cells, which are responsible for producing autoantibodies [124,126,127,128]. Tun-Kyi et al. reported the role of Pin1 in Toll-like receptor (TLR) signaling and type 1 IFN activation using bone marrow-derived myeloid dendritic cells from mice in 2011 [129]. Pin1 is activated by the activation of TLR7 and TLR9 in response to foreign DNA signatures. Pin1 then binds to IL-1 receptor-associated kinase-1 (IRAK1), inducing conformational changes that promote dissociation from the Myddosome receptor complex and its activation. This facilitates the activation of the transcription factor Interferon Regulatory Factor 7 (IRF7) and induces the production of type 1 IFN. Wei et al. connected the link between Pin1-mediated production of type 1 IFN via TLR-7/TLR-9/IRAK-1/IRF-7 signaling and the development of SLE in 2016 [130]. They observed that Pin1 activity is elevated in monocytes from SLE patients compared to control groups. Furthermore, in vitro Pin1 knockdown, or treatment of cells with Pin1 inhibitor All-*trans* retinoic acid (ATRA), suppresses Pin1/IRAK-1/IRK-7. In addition, ATRA treatment of a lupus-prone mice model significantly reduced SLE-related phenotypes. Therefore, targeting Pin1-mediated production of type 1 IFN via TLR-7/TLR-9/IRAK-1/IRF-7 signaling may provide a promising novel therapeutic approach for SLE treatment and further studies are clearly warranted.

In addition, recent research has identified Pin1 as an important regulator of IL-6 expression in SLE [131]. IL-6 is a pro-inflammatory cytokine that plays a key role in the differentiation of B cells into antibody-producing plasma cells, differentiation of CD4 naïve T-cells into Th17 cells, and differentiation of CD8+ T-cells into cytotoxic T-cells [132]. Dysregulation of IL-6 production can result in autoantibody production, contributing to SLE by promoting the survival and differentiation of autoreactive B cells, which produce antibodies that target self-tissues. IL-6 expression blockade greatly improved the phenotype in SLE [133]. IL-6 signaling is mediated by the IL-6 receptor (IL-6R) and the signal transducer and activator of transcription 3 (STAT3), and dysregulation of this pathway has been implicated in the pathogenesis of several diseases, including cancer and autoimmune disorders. Pin1 interacts with STAT3, promoting its phosphorylation and nuclear localization [134]. Pin1 inhibition in Juglone-treated NZB/W F1 lupus mice resulted in suppression of B-cell differentiation and T-cell activation compared to non-treated mice. Furthermore, decreased STAT3 phosphorylation was observed in T-cells of Juglone-treated mice [131]. These findings suggest that Pin1 plays a critical role in regulating IL-6 signaling and may contribute to the pathogenesis of SLE. 

As well as multiple sclerosis and systemic lupus erythematosus, CypA has been found to be associated with rheumatoid arthritis (RA), which is characterized by chronic inflammation of joints that leads to severe pain, swelling, and stiffness of joints. The chronic inflammation can gradually erode the cartilage and bone within the joints, causing deformity, and eventually loss of function of the affected joints. The secreted form of CypA is found to be elevated in the synovial fluids of rheumatoid arthritis patients and macrophages in the synovial lining layer are the major source of this CypA [135,136]. CypA contributes to the destruction of cartilage and bone by upregulating MMP-9 (Matrix Metalloproteinase-9) expression and adhesion of monocytes/macrophages to extracellular matrix [137]. CypA also affects IL-8-directed chemotaxis in neutrophils of RA patients and is responsible for CypA-mediated neutrophil migration into the joints, elevated MMPs secretion, and cell invasion of synoviocytes, which are key pathological mechanisms of RA [138,139,140,141].

### 5.2. Cancer

Cancer is a highly intricate and constantly evolving disease that arises as a result of often multifactorial interactions between genomic, epigenomic, proteomic, and environmental factors. It represents a complex and heterogeneous class of disorders characterized by a diverse array of molecular and cellular aberrations, including dysregulated proliferation, impaired cell death, genomic instability, and immune evasion. Since proline isomerization mediated by prolyl isomerases is a critical post-translational modification that regulates protein folding, stability, and function, emerging evidence suggests that aberrant expression of prolyl isomerases can contribute to the pathogenesis of cancer by affecting a wide range of oncogenic signaling pathways, including DNA repair, cell cycle regulation, apoptosis, and immune surveillance. A large-scale human tissue microarray study compared normal and cancerous tissues to reveal that the protein expression level of Pin1 is significantly overexpressed in most common cancers including prostate, breast, lung, and colon cancer [142]. Studies revealed overexpression of Pin1 in both mRNA and protein levels for oral squamous cell carcinoma [143,144], gastric cancer [145], and acute myeloid leukemia [146], and only at the protein level in colorectal cancer [147,148], osteosarcoma [149], and pancreatic cancer [150]. Recently in 2023, Naito et al. reported that Par14 is overexpressed in prostate cancer, and it promotes proliferation by directly binding to the androgen receptor and regulating androgen receptor signaling pathways [151]. On the other hand, FKBPs (mainly FKBP51 and FKBP52) and Cyp40 are overexpressed in steroid hormone-dependent cancers such as breast and prostate cancer [152,153,154]. Recently, a few groups have reported overexpression of FKBP10, FKBP11, FKBP5 in renal cancer and hepatocellular carcinoma [155,156,157].

Overexpression of Pin1 has been correlated with the poor clinical outcome of cancer patients by affecting multiple oncogenic signaling pathways [158,159,160]. Pin1 was first described as a mitotic regulator of cyclin D1 in HeLa cells [161]; overexpression of Pin1 increases the cyclin D1 level by multiple signaling pathways including C-Jun/Fos, β-catenin, and NF-Kβ [161,162,163]. Moreover, Pin1 directly binds and activates 56 oncogenic proteins and inactivates 26 tumor suppressors [164]. The *cis*/*trans* isomerization also stabilizes and increases the half-life of oncogenic proteins by preventing their nuclear localization and degradation via the ubiquitination–proteasome pathway [79]. Pin1 knockout and/or knockdown induces apoptosis, and suppression of oncogenesis both in vitro and in vivo [165,166,167,168,169]. The rs2233678 polymorphism in the Pin1 promoter region (-842 G > C) has been shown to decrease the risk of cancers including breast cancer [170,171]. There are 32 somatic mutations reported in the Pin1 gene among different types of cancer suggesting a correlation between Pin1 mutation and the increased risk of cancer [80]. Additionally, Pin1 also plays a role in centrosome amplification and genomic stability [172]. Pin1 mediates tamoxifen resistance in the MCF-7 breast cancer cell line by increasing the mRNA and protein expression of LC-3, a marker for autophagy, which is believed to be utilized by cancer cells as a coping scheme for survival at the point of drug selection [173]. Pin1-dependent vascular endothelial growth factor (VEGF) production in MCF-7 cells was observed in tamoxifen resistant cells [174]. Wang et al. showed that Pin1 mediated cisplatin resistance in cervical cancer via the FOXM1/WNT-β-catenin signaling pathway in 2016 [175]. In addition, some studies revealed new insights regarding Pin1 as a potential regulator for induction and maintenance of pluripotent stem cells via the regulation of Oct4, which is an imperative transcription factor for embryonic stem cells [176]. Despite most studies suggesting Pin1 is pro-cancerous, it has also been observed that Pin1’s high expression correlates with a better prognosis in melanoma, prostate, and testis cancer patients, as revealed from a bioinformatics analysis of tumors in the Human Protein Atlas [87]. They suggested that the relationship between p53 and Pin1 expression in cancer cells may explain this phenotype. Pin1-mediated isomerization of wild-type p53 can potentially activate its tumor-suppressive functions, whereas mutant p53, which is more prevalent in cancer, may activate its oncogenic function. Taken together, the above findings suggest that Pin1’s function in cancer cells can be complicated and whether Pin1 is pro or against cancer cell growth depends on the tissue and the existence and status of other protein factors. Therefore, future research delineating Pin1’s function in different cell types and genetic background is greatly needed and would be important to determine Pin1 as a target for cancer therapy, which is also highly relevant to precision medicine.

FKBPs, such as FKBP5, FKBP51, and FKBP52, are important components of steroid hormone receptor-Hsp90 complex [152,177]. They directly bind to Hsp90, via their TPR domains, and serve as co-chaperones [178]. FKBP52 binds to androgen receptor (AR) [179], estrogen receptor α (ERα) [154], glucocorticoid receptor (GR) [180], and progesterone receptor (PR) [181] in the nucleus. Maeda et al. reported that FKBP52 regulates dimerization of AR in 2022 [182]. Similar to Pin1, these interactions lead to increased transcriptional activity of these receptors, which leads to cell proliferation in breast and prostate cancer (Figure 5B). FKBP51 also mediates radio-resistance in malignant melanomas via regulation of the NF-Kβ pathway [183]. 

Cyp40 can also bind to the Hsp90-ERα complex and increase the transcriptional activity of the receptor [184]. Additionally, Cyp40 siRNA experiments in ALK+ ALCL cell lines directed the potential role of Cyp40 in oncogenesis in lymphoma. The role of PPIases in cancer has been reviewed by several other authors and it is recommended to refer to them for more in-depth information [81,178,185,186].

### 5.3. Infectious Disease

Prolyl isomerases have been known to play a significant role in various stages of viral propagation. HIV-1 (Human Immunodeficiency Virus type 1) provided the most prominent example of proline isomerization in viral pathogenesis for the first time. Two distinct PPIases, human CypA and Pin1, have been found to play an important role in the life cycle of HIV and induce viral pathological effects in humans. The mature HIV-1 capsid is a fullerene cone-shaped structure composed of 125 hexametric and 12 pentameric units. Hexametric units consist of 6 p24 molecules arranged in a circular ring, stacked on top of each other to form a lattice structure that makes up the bulk of the capsid, while pentameric units consist of 5 p24 molecules arranged in a pentagonal ring, and they are located at the narrow end of the cone and stabilize the overall structure [187]. The capsid core is responsible for aiding viral infection by interacting with host cells, transporting the viral genome and the reverse transcriptase machinery to the host nucleus [188]. Since the first evidence of interaction between CypA and HIV-1 was found [189], several independent groups have shown that the incorporation of CypA into HIV particles facilitates HIV replication, maturation, and infectivity [190,191,192]. The underlying mechanism has been deciphered by the collective works using both crystallography and NMR spectroscopy [45,189,193]. When the virus is transported to the host cell, CypA binds to the capsid and catalyzes proline isomerization at the Gly89-Pro90 site in the capsid protein, leading to conformational changes that increase the affinity between CypA and capsid and the formation of a stable complex that is important for unpacking the capsid (Figure 5C). The N-terminal domain of the HIV-1 capsid (residues 1-151) has been captured complexed with CypA by X-ray crystallography [45] (PDB: 1AK4). Specifically, the active site of CypA was found to bind to a flexible and exposed loop region (residues 85-93) of the capsid protein located between helix 4 and 5 on the N-terminal domain. Additionally, it was found that the CypA-capsid interaction also prevents the binding of and replication restriction from human TRIM5α. Genetic tools and functional assays have shown that disrupting the CypA–capsid interaction can render HIV-1 susceptible to endogenous TRIM5α [194].

In addition to CypA, Pin1 also plays a significant role in facilitating the life cycle of HIV-1 and promoting infection. Pin1 is involved in several key processes including capsid core uncoating, reverse transcription, and integration. In 2010, Misumi et al. reported that Pin1 interacts with the HIV capsid and the proline isomerization is essential for capsid–capsid protein dissociation and disassembly of the HIV-1 core into the host cells at the early stage of the HIV-1 life cycle [195]. Using Pin1 pull-down assay, they showed that Pin1 binds preferentially at the pSer16-Pro17 motif in HIV-1 capsid proteins. The results indicate that the capsid core, which has been phosphorylated within the viral particle following its release from infected cells, binds directly to Pin1 upon entering host cells and then undergoes an uncoating process. Furthermore, restrictions in reverse transcription and HIV-1 replication were observed when a double mutation S16A/P17A was introduced to the capsid. Additionally, suppressing Pin1 expression led to the attenuation of HIV-1 replication and resulted in the accumulation of capsid core particulates in the cytosol of the host cells.

In addition to its role in promoting HIV uncoating and replication, Pin1 also binds and inhibits APOBEC3G, a cytidine deaminase protein that has antiviral activity, by restricting HIV-1 replication, leading to its degradation via proteasome [196]. This degradation mechanism was originally thought to be mediated solely by HIV-1’s viral infectivity factor (Vif), a viral protein [197]. Therefore, Pin1-mediated depletion of APOBEC3G represents a novel mechanism of its degradation. Pin1 has been shown to promote viral genome integration into the host cell genome. This is mediated by the JNK-mediated phosphorylation of the Ser57-Pro58 motif of HIV-integrase via the NF-Kβ signaling pathway [46,198]. 

In comparison to cyclophilins and Pin1, the role of FKBPs in HIV infection is not as clear. However, recently, CRISPR-Cas9 screening and co-immunoprecipitation assay showed that FKBP3 indirectly binds with HIV-1-long terminal repeats (HIV-LTR) and through YY1 and HDAC 1/2 proteins [199]. These interactions allow histone deacetylation of HIV-LTR which is shown to be associated with inhibition of HIV replication and promotion of HIV-1 latency.

Apart from HIV, hepatitis C virus (HCV) is the most widely studied viral model system to understand the mechanism of viral replication. The insight into the role of cyclophilins in HCV replication came from in vitro cyclosporin inhibition of HCV replication [200,201]. It was demonstrated that CypA and CypB bind with HCV NS5A/B, an RNA-dependent RNA polymerase, promoting its RNA-binding activity and thereby HCV replication [202,203]. Although the crystal structure of CypA/B with NSB5A/B has not yet been solved, Li et al. predicted their complex structure using protein–protein docking and molecular dynamics simulation and suggested in 2022 that the hydrophobic pocket (Ile57, Phe60, Trp105, Trp121) of CypA interacts with the proline-rich D2 domain of NS5A. The exact mechanism underlying the isomerization and hard evidence depicting the complex structure of CypA-NS5A has yet to be elucidated. Interestingly, Pin1 has also been shown to bind directly to both NS5A and NS5B and to modulate HCV replication [204]. In addition to cyclophilins and Pin1, FKBP, especially FKBP8, has also been shown to bind with NS5A of HCV via its TPR domain by immunoprecipitation assay, which promotes HCV replication [205]. 

As well as HIV and HCV, there is evidence to show PPIases also impact on hepatitis B virus (HBV) replication and virion secretion. Pin1 stabilizes the hepatitis B virus by directly binding to two separate proteins, HBV core protein and HBV X protein. These proteins are vital for maintaining HBV genome integrity and replication. Dephosphorylation at the sites Thr160 and Ser162 of HBV core protein results in the loss of Pin1 binding and degradation of HBV core protein by lysosome [206]. Pin1 can also bind to the highly conserved Ser41-Pro42 sequence of HBV X and stabilize the protein, which contributes to viral replication and viral entry into the host cells [207,208]. Interestingly, this interaction has also been indicated to play a role in virus-induced tumorigenesis, i.e., hepatocarcinogenesis [206]. Furthermore, Par14 and Par17 can also promote HBV virus replication through formation of a ternary complex with covalently closed circular DNA and HBX protein [209]. As well as Pin1, there is limited information on FKBPs in relation to HBV, but there is some indirect evidence suggesting the involvement of cyclophilins. Using cyclophilin inhibitors and RNA interference experiments, several groups have shown that CypA directly binds to HBX [210], and promotes HBV replication, production and secretion [211,212,213]. Finally, to conclude this section, the connection between PPIases and viral infections covered here is far from being complete and one can refer to additional review articles by Yamamoto et al. [214], Wu et al. [215] and Kanna et al. [216] to access more comprehensive details regarding other viral infections, including SARS-CoV. 

### 5.4. Neurodegenerative Disease

Neurodegenerative diseases, such as Alzheimer’s disease (AD) and Parkinson’s disease (PD), are primarily characterized by the progressive loss of neurons. Prolyl isomerases are expressed at higher levels in neurons than in other differentiated cells. Numerous studies have demonstrated the protective role of proline isomerization during neuronal differentiation. Consequently, prolyl isomerase dysregulation has been associated with abnormal aggregation of neuronal substrates, including tau, amyloid precursor proteins (APP), and α-synuclein in age-dependent neurodegenerative diseases. 

In Alzheimer’s disease, the accumulation of proteins such as tau and APP into plaques is believed to be partly caused by changes in proline isomerization. The *cis*/*trans* isomerization of the pThr231-Pro232 motif in tau mediated by Pin1 plays a crucial role in the development and progression of AD and tauopathies [217] (Figure 5D). *Trans*-tau promotes the health of neurons, while the *cis*-isomer of tau has been linked to AD-like symptoms, including decreased microtubule formation, increased tau phosphorylation, aggregation, and tangle formation which led to neuronal apoptosis, and neurodegeneration [218]. Pin1 regulates tau by converting *cis*-tau to *trans*-tau, which then binds to microtubules and maintains their dynamics. Additionally, Pin1-deficient mice have been found to display tau pathology and neuronal loss. In addition to tau, Pin1-mediated isomerization of the APP protein has also been linked to AD. Pin1 binds and catalyzes the pThr668-Pro669 motif in APP [219]. Isomerization from *cis*- to *trans*-APP results in the stabilization of APP in plasma membrane and decreased protein turnover of APP, which leads to a decrease in the activity of Glycogen Synthase Kinase 3 Beta (GSK3β), a detrimental factor in neuronal death, and inhibition of APP binding to Fe65, which modulates trafficking and proteolytic processing and production of β-amyloid peptides that lead to neuronal apoptosis when binding to the intracellular domain of APP [220]. In contrast, the *cis*-isomer of APP promotes an increase in β-amyloid peptide production, resulting in plaque formation and the amyloidogenic processing pathway [221]. This effect is also observed when Pin1 levels are low in neurons. Altogether, this evidence indicates the protective role of Pin1 in AD.

Parkinson’s disease (PD) has been linked to the pathogenesis of α-synuclein, as indicated by the presence of amyloid-like aggregates of α-synuclein in the brains of PD patients. In addition to its role in AD, Pin1 has also been found to accumulate and co-localize with α-synuclein inclusions in the Lewy bodies of human PD brains [222] (Figure 5D). Pin1 promotes the formation of α-synuclein inclusions by stabilizing α-synuclein and enhancing its binding with synphilin-1. In addition to Pin1, it has been shown that CypA binds and catalyzes the proline isomerization of Pro128 located at the C-terminus of α-synuclein, resulting in misfolding and aggregation [223]. Moreover, FKBP12 has also been associated with α-synuclein aggregation [224]. More detailed information on the role of Pin1 in neuronal signaling can be found in the review by Fagiani et al. in 2021 [225].

## 6. Prolyl Isomerase Inhibitors

As discussed above, PPIases and their mediated proline isomerization are involved in the pathogenesis of many diseases. It is therefore no surprise that PPIases have been recognized as potential therapeutic targets for disease treatments. The mode of action of different PPIase inhibitors varies depending on the type of PPIase and the specific compound. In this section, we provide a concise overview of various inhibitors currently characterized for each type of PPIase using numerous methods such as structure-based drug design, natural product screening, and hit-to-lead optimization. Table 2 summarizes the inhibitors discussed for quick reference. Achievements of the use of PPIase inhibitors in preclinical and clinical settings to explore their anti-inflammatory, anti-viral, and anti-cancer properties will be discussed. 

### 6.1. Cyclophilin Inhibitors

Human CypA is the most well studied cyclophilin, and cyclosporine is a well-characterized inhibitor that binds to CypA with high affinity as shown by cell-free competitive Enzyme-linked immunoassay (ELISA) experiments (Kd = 6 nM, IC_50_ = 420 ± 56 nM) [245]. Cyclosporine is a cyclic undecapeptide, originally isolated from ascomycete fungi, *Tolypocladium inflatum,* by Borel and colleagues in 1976 [246] (Figure 6A). It binds to the hydrophobic core in the active site of the PPIase domain of cyclophilins, and the resulting complex ends up binding and inhibiting calcineurin. Cyclosporine, therefore, is a calcineurin inhibitor and is commonly used as an immunosuppressant agent that suppresses T-cell activity by inhibiting calcium-dependent IL-2 production [247].

Interestingly, although high doses of cyclosporine (≥4–5 mg/kg per day) are recommended for organ transplantation and this dose is required for immune suppression, several independent groups have shown that low doses of cyclosporine (≤3 mg/kg per day) stimulate pro-inflammatory cytokines such as IL-12, IFN-gamma, and TNF-α and promote cell death in mouse models [248,249,250]. The phase I/II clinical trial conducted by Ross et al. in 1997 also suggests increased survival in advanced non-small cell lung cancer patients who were given low dose of cyclosporine (1–2 mg/kg per day) [251]. The contrasting role of low doses of cyclosporine is favorable for its use as anticancer therapeutics, although our understanding of this phenomena is still limited. 

In principle, cyclosporine holds promise as an antiviral drug by targeting host cyclophilin, which is required for the replication of some viruses, including HIV and hepatitis B and C viruses. Indeed, in vitro experiments showed that cyclosporine inhibits HIV replication and initial clinical trials of cyclosporine in HIV patients showed a decrease in the viral load. However, using cyclosporine for the treatment of HIV is challenged by its immunosuppressive activity that worsened the clinical condition of the immunocompromised HIV patients. As a result, cyclosporine is not primarily used for the treatment of HIV as of today but is used in combination therapy with antiretroviral drugs in low doses (0.3–0.6 mg/kg), which limits its prospects as shown in a pilot clinical trial with 20 HIV-infected patients [252]. To improve the clinical outcome, several analogues of cyclosporine were developed (Figure 6). Debio-025 or alisporivir is one of them that does not have immunosuppressive activity but inhibits cyclophilin more potently and selectively than cyclosporine. They showed promising results in an antiviral assay against HIV [227] and HCV [228]. An in vitro experiment showed clearance of the HCV replicon when Debio-025 was used in combination with ribavirin or HCV NS3 protease or NS5B polymerase inhibitors [253]. Subsequently, it became the first cyclosporine analog entered for clinical trial, which showed a decreased viral load with 23 HIV-HCV co-infected patients [254]. The drug then moved to phase II trial with lower doses and in combination with pegylated interferon on 90 chronic HCV-infected patients which also significantly reduced the viral load [255]. However, in both clinical trials, reversible hyperbilirubinemia was reported. In addition, a clinical trial on 290 chronic HCV genotype 1 patients was carried out using Debio-025 with pegylated interferon and ribavirin; however, the result has yet to be released (Debiopharm International SA 2009–2016, NCT00854802). SCY-635 is another non-immunosuppressive cyclosporine analogue that possesses potent inhibitory effects on HCV RNA replication. SCY-635 exhibited particularly increased potency when used in combination with α-interferon and ribavirin, the standard-of-care treatment for HCV infections [231]. SCY-635 has been tested in a phase I trial and no toxicity effects were observed. Although no further report on its clinical use is available, SCY-635 is a promising molecule to enter the next stage of clinical trial. NIM811 is also a non-immunosuppressive analog with increased antiviral activity against both HIV and HCV infection [256,257]. It has entered Phase II clinical trials and showed synergistic antiviral activity against HCV when used in combination with α interferon [229]. Additionally, along with another cyclosporine analog, UNIL025, it exhibited neuroprotection effects because these drugs inhibit mitochondrial permeability transition and prevent calcium-induced brain mitochondria swellings [258]. CRV431 (formerly known as CPI-431-32) is probably the most potent non-immunosuppressive cyclosporine analog with broad-spectrum antiviral activity against HIV, HCV, and HBV. Using an in vitro co-culture model for HCV and HIV co-infection, CRV431 inhibited replication of HCV (IC50 = 0.18 ± 0.03 nM) and HIV-1 (IC50 = 0.23 ± 0.01 nM). It prevented the CypA–capsid and CypA–NS5A interactions in HIV-1 and HCV, respectively, rendering the virus non-infectious due to premature uncoating [259]. Gallay et al. tested the efficacy of CRV431 in transgenic mice and observed that the drug inhibited liver HBV DNA and cleared HBsAg serum level with no known toxicities in 2019 [212]. Interestingly, CRV431 decreased fibrosis and showed antitumor activity in a chronic liver disease mouse model. Due to its antisteatotic effects, a phase II clinical trial on 47 participants with non-alcoholic fatty disease has recently been completed, although the result is not yet available (Hepion Pharmaceuticals, Inc).

In addition to cyclosporine, sanglifehrins are a different class of naturally occurring cyclophilin inhibitors which were isolated and discovered by Sanglier et al. from actinomycetes, *Streptomyces* sp. A92-308110, in 1999 [260] (Figure 6B). There have been about 20 sanglifehrins isolated so far and Sanglifehrin A (SFA) is the most abundant compound [226]. SFA exhibits strong binding to CypA (IC50 = 6.9 ± 0.9 nM) with immunosuppressive activity. In contrast to cyclosporine, SFA blocks IL-2 mediated T-cell proliferation at the G1 phase of the cell cycle in a calcium-independent manner without interfering with IL-2 transcription [245,261]. Furthermore, Pua et al. revealed that SFA achieves this by binding to and inhibiting the activity of inosine 5′-monophosphate dehydrogenase 2 (IMPDH2), a known regulator of cell growth in 2017 [262]. Sanglifehrins A, B, C and D have been reported to inhibit HCV replication in vitro and in vivo more potently than cyclosporine [263,264]. In addition to antiviral activity, SFA has also been characterized as an anti-fibrotic agent and a dendritic cell chemokine and migration inhibitor [265,266]. With the objective of enhancing the binding affinity and specificity of the original sanglifehrin compounds for CypA, several groups have been working to develop sanglifehrin analogs. In a study by Bobardt et al. [267], it was demonstrated that NV556, an analog of sanglifehrin, exhibited antiviral effects against HCV infection in a humanized liver mice model. They found that a single oral dose of NV556 effectively suppressed viral load and prevented viral relapse for up to nine months. Additionally, NV556 displayed the ability to reduce liver fibrosis and hepatocellular carcinoma in mouse models of non-alcoholic steatohepatitis [230,233]. NV651 is another sanglifehrin analog which has demonstrated efficacy in reducing cancer cell proliferation in both in vitro and in vivo studies. Furthermore, it is more potent compared to Sorafenib, an FDA approved drug and the current first-line treatment for late-stage hepatocellular carcinoma, in HEPG2 cells (IC50_NV651_ = 6.3 nM, IC50_Sorafenib_ = 2331 nM) [232]. NV651 shows promising potential as an anti-HCV therapy, particularly as patients’ resistance to Sorafenib is developed. An in vitro experiment using a combination of NV651, and cisplatin inhibited cell proliferation and increased apoptosis. Additionally, impaired cell cycle progression and DNA repair was observed in HEPG2 cells with combination therapy compared to cisplatin alone [268]. At the time of writing this review, clinical trials for sanglifehrins and its analogues have not yet been initiated. 

### 6.2. FKBP Inhibitors 

FK506 and rapamycin are two FDA approved natural ligands for immunosuppression. Although the targets of these drugs were unknown when they were first discovered, they were found to bind to FKBPs, especially FKBP12 [269,270,271,272]. These macrolides are obtained from actinomycetes, *Streptomyces tsukubaensis*, and *Streptomyces hygroscopicus*, respectively (Figure 7). The functional group of both FK506 and rapamycin is the pipecolic acid group. Studies showed that the pipecolate (piperidine-2-carboxylic acid) moiety of these drugs binds to the active site of FKBPs, as revealed by the crystal structure of FK506 and rapamycin complexed with FKBP12, respectively [273,274,275]. The complexes inhibit calcineurin and mTOR, respectively, leading to the inhibition of their downstream effectors. Further studies showed that they also have neuro-regenerative and neurotrophic effects, suggesting their potential use for the treatment of neurodegenerative diseases such as Alzheimer’s disease and Parkinson’s disease. One major challenge in using these drugs for these diseases was their immunosuppressive effects. As a matter of course, tremendous efforts to develop non-immunosuppressive analogs of FK506 and rapamycin were made. Several research groups developed strategies to modify the chemical groups of these ligands. One strategy is to keep the pipecolate group while modifying the moieties that interact with calcineurin and FKBP-rapamycin-binding (FRB) domain of mTOR to some other aliphatic chain including the occasional bulky phenyl group. Alternatively, the esters of the pipecolate group are modified into amidyl, sulfonyl, or carbonyl oxygens [61]. This approach guided by structure-activity relationship studies ultimately led to new small molecule inhibitors for FKBPs. GPI-1485 was one of the first synthetic FKBP inhibitors to demonstrate a successful design in its development (Figure 7). It is an inhibitor for FKBP12 that entered Phase II clinical trials in the treatment of Parkinson’s disease where it showed marginal effects in compared to a placebo (NINDS NET-PD Investigators, 2007). Furthermore, several groups refuted the application of GPI-derivative compounds as neurotrophic and neuroprotective agents for the treatment of Parkinson’s disease because these compounds bind poorly to FKBP12, opposing the earlier observations [276,277]. They reported that GPI compounds exhibited lower levels of inhibition on FKBP12 compared to FK506 and rapamycin. Additionally, these compounds were observed to undergo ester hydrolysis, which impacted their activity, half-life, and bioavailability in the neurons of the dorsal root ganglion. As well as neurodegenerative diseases, FKBP inhibitors have also shown good potential in treating microbial infections [278,279,280,281,282], and cancer [283,284,285] and new research is greatly needed to back up repurposing these drugs or inhibitors for other diseases. 

While the investigation of currently available drugs began with FKBP12, there is increasing interest in studying the roles of FKBP51 and FKBP52 in the pathogenesis of various diseases and in targeting them for drug discovery. Designing inhibitors that specifically target one of the FKBP homologs has proven to be challenging, given the high degree of similarity in the catalytic domains of proteins such as FKBP51 and FKBP52. Recent developments of SaFit1 and SaFit2 (Selective Antagonist of FKBP51 by induced fit) have shown increased potency (Ki = 4 nM and 8 nM, respectively) and selectivity to the FKBP51 isoform [234] (Figure 7). Meanwhile, more research is crucial to characterize the PPIase activity and biological function of these FKBP homologs. For a more thorough review of FKBP inhibitors, reading the review by Kolos et al. [61] is highly recommended.

### 6.3. Pin1 Inhibitors 

Among all Parvulins, Pin1 is the only extensively studied one, and the development of its inhibitors has been intensely pursued due to its involvement in diseases. In particular, overexpression of Pin1 has been linked with tumorigenesis in several types of cancer. Therefore, most of the currently available inhibitors have been characterized as anticancer agents in various cancer cell lines and mouse models, with the primary aim of suppressing tumor growth and/or preventing tumor recurrence, as well as enhancing the understanding of Pin1 biology. Pin1 inhibitors can be divided into covalent (Figure 8) and noncovalent inhibitors (Figure 9), both of which have pros and cons. Covalent inhibitors form covalent bonds with side chain residues in Pin1 and permanently modify it, and they can have high potency and longer duration of action. The noncovalent inhibitors bind reversibly with Pin1 through binding, unbinding, and rebinding, on the other hand, allowing more precise control over the inhibitory effects [238]. 

The first Pin1 covalent inhibitor, Juglone, also known as 5-hydroxy-1,4-napthoquinone, was isolated from walnuts and characterized more than two decades ago [286]. Two Juglone molecules covalently bind to the side chains of Cys41 and Cys69 in the Pin1 catalytic domain simultaneously, resulting in its partial unfolding, loss of activity, and further degradation by proteases [286,287]. Since its discovery, Juglone and its derivatives have demonstrated significant anticancer properties, including antiproliferative [288], anti-angiogenic [289], pro-apoptotic [290], and inhibitory effects on migration and invasion [291,292]. These compounds have been extensively investigated as potential cancer therapeutics in various types of cancer, including breast cancer [293,294], cervical cancer [291,295], lung cancer [235], prostate cancer [296,297], glioma [288,289,298], pancreatic cancer [292,299] and bladder cancer [300,301]. However, clinical studies of Juglone have been unsuccessful due to its cytotoxicity, poor solubility, and stability in biological systems. The cytotoxicity could come from off-target effects of Juglone due to its lack of structural complexity and therefore limited specificity, although Juglone has been shown to be selective towards Pin1 without affecting other PPIases such as FKBPs and cyclophilins. Nevertheless, the action mechanism and drug properties of these derivatives are yet to be understood. KPT-6566, a promising covalent inhibitor with added functional groups to the naphthyl backbone, exhibits greater specificity than Juglone (Figure 8). Molecular docking suggests that the electrophilic sulfanyl–acetate group may form a disulfide bond with Cys113, a highly conserved residue in the catalytic active site in Pin1, leading to its degradation and the production of reactive oxygen species (ROS) and DNA damage. Furthermore, KPT-6566 demonstrates efficient cell permeability and inhibitory effects on Pin1 function, both in vitro and in vivo, as evidenced by its ability to suppress lung metastasis in a breast cancer in mouse model [236]. KPT-6566 holds promise in the treatment of various cancers such as breast, pancreas, prostate, and lung cancers, due to its ability to exhibit antiproliferative and proapoptotic effects [164]. A recent study showed the therapeutic importance of Juglone and KPT-6566 as a potent anticancer drug for CD44+ CD133+ colorectal tumor-initiating cancer cells [302]. Recently, some other Juglone derivatives with triazolyl group and electron withdrawing groups such as –NO2 and -CN have been shown to be more selective to cancer cells than Juglone and can be potential anticancer agents [303]. Nevertheless, the action mechanism and drug properties of these derivatives are yet to be understood.

In addition to KPT-6566, (S)-2 is another covalent inhibitor that has been shown to inhibit Pin1 by covalently binding to Cys113, although they do not share structural similarity. An in vitro protease-coupled assay on the prostate cancer cell line PC-3 showed that (S)-2 inhibited Pin1 and exhibited moderate cytotoxicity via suppression of CyclinD1 expression [237]. BJP-06-005-3, a compound derived from the non-covalent inhibitor D-PEPTIDE, utilizes a chloroacetamide group to act as an electrophile and forms a covalent bond with Cys113 in Pin1. BJP-06-005-3 promotes Pin1 degradation, leading to a downstream suppression of Pin1-mediated Myc-Ras signaling pathways and reduced cell viability in pancreatic ductal adenocarcinoma [238]. Recently, a more potent covalent inhibitor of Pin1 called Sulfopin, which also acts on Cys113, has been discovered by Dubiella et al. in 2021 [240]. Two independent chemoproteomics methods showed that Sulfopin is highly selective for Pin1. Furthermore, they also showed that inhibition of Pin1 activity suppressed the downstream c-Myc target genes and reduced tumor progression in both in vitro and in vivo neuroblastoma and pancreatic cancer models. In addition, more recently, through structure-based optimization, Liu et al. revealed another novel covalent inhibitor of Pin1, ZL-Pin13 (IC50 = 0.067 ± 0.03 uM) in 2022 [239]. It covalently binds with Cys113 of Pin1 and selectively inhibits Pin1 as determined by X-ray crystallography. In vitro studies with MDA-MD-231 cells showed that ZL-Pin13 inhibited proliferation of TNBC cells as well as downregulated Pin1-substrates c-Myc, MCL-1 and Cyclin D1 [239].

Regarding non-covalent inhibitors for Pin1, D-PEPTIDE (Ac-Phe-D-PhosThr-Pip-Nal-Gln-NH2) is a non-natural peptide-based non-covalent inhibitor of Pin1 that binds to the active site of Pin1 with high specificity (Figure 9). X-ray crystallography demonstrated that it binds at the active site of Pin1 and interacts with Lys63 and Arg69 with its phosphate moiety [55]. The non-natural proline analogue Pipecolic acid (Pip), with its six-membered ring, is responsible for the inhibitor’s high specificity for Pin1. An in vitro isothermal titration calorimetry experiment showed an improved potency and specificity of D-PEPTIDE (Ki = 20.4 ± 4.3 nM). Despite the challenge of cell permeability due to the phosphate group, this discovery paved the way for the idea that incorporating a cyclic peptide polyketide bridge resembling FK506 and rapamycin into Pin1 inhibitors’ structure could significantly enhance their binding affinity [55].

In addition to D-PEPTIDE, there are other non-covalent inhibitors primarily targeting the prolyl isomerase domain of Pin1, including peptidyl [304], and small molecules such as aryl indanyl ketones [305], selenium derivatives [306], non-heterocyclic derivatives [307], pyrimidine derivatives [308], and thiazole derivatives [309]. Pfizer made a significant effort to produce a non-covalent inhibitor by developing three small molecule inhibitors named Compound **21b**, Compound **23b**, and Compound **22c** (racemic), using a combination of structure-based drug designs and combinatorial methods [241,242,243]. While compound **21b** contained a phosphate group and showed high potency, it was not active in cells. Replacing the phosphate group with a carboxylic acid in compound **23b** led to a decrease in both potency and activity in cells. However, through structure-based optimization and by exploring H-bond interactions in Pin1’s active site, they developed Compound **22c** (racemic), a more potent drug with increased cell permeability. In addition, recent studies utilizing in silico receptor-based pharmacophore modeling, consensus docking, and molecular dynamics simulations coupled with in vitro cell-based assays discovered two non-covalent Pin1 inhibitors, VS1 and VS2. They both inhibit Pin1 activity, and VS2 in particular showed antiproliferative activity with IC50 to be 19 ± 310 μM, 66 ± 6 μM, and 29 ± 4 μM in three ovarian cancer cells, SKOV3, OVCAR5 and OVCAR3, respectively, highlighting the role of powerful computational screening algorithms in the development of new potent Pin1 inhibitors [244].

Despite all these efforts, there has been no compound targeting Pin1 to enter clinical trials, as of writing this paper. However, in a drug-repurposing effort, ATRA, an FDA-approved drug for the treatment of acute promyelocytic leukemia (APL), has also been shown to be a non-covalent Pin1 inhibitor. ATRA inhibits Pin1 by binding to its active site and disrupts the interactions between Pin1 and its substrate proteins and induces its degradation. ATRA has been tested in clinical settings for various types of cancers, including metastatic breast cancer in phase II trials [310], non-small cell lung cancer in phase II trials in combination with paclitaxel and cisplatin [311], pancreatic cancer in phase I trials in combination with gemcitabine-nab-paclitaxel [312], and adenoid cystic carcinoma in phase II trials [313]. ATRA demonstrated enhanced responses and survival rates in all clinical trials, while also being well-tolerated and safe. Therefore, ATRA has been the only Pin1 inhibitor that has moved to clinical trials so far and is ready to progress to subsequent phases of clinical trials. 

## 7. Conclusions 

As evidenced above, the discovery and study of proline isomerization in protein folding and regulation has opened up exciting avenues in understanding the fundamental disease mechanisms and significant progress has been made in applying these findings to yield potential therapeutics that can have antiviral, anticancer, or neurotrophic/neuroprotective properties. Although much is known about the biological functions of CypA, FKBP12, FKBP51/52, and Pin1, other isoforms and closely related proteins with structural and functional similarities are still poorly understood and require further study. These studies are not only crucial for establishing attractive protein targets for drug discovery and development but will also help to reach higher selectivity and specificity in the inhibitors to achieve the required efficacy and safety profiles for clinical testing.

Among all these PPIase inhibitors, cyclophilin inhibitors appear to be the most successful, with a few compounds having moved into clinical testing. Comparatively, inhibitors targeting FKBP and Pin1 are relatively less developed. The reason could be that targeting FKBP and Pin1 is more complicated due to their essential and heterogenous functions in different tissues such as in neurons and cancer cells, complex network of interacting proteins and pathways being affected, and inhibitors not being specific to the isoform. Tissue-specific drug delivery strategies, such as prodrug or soft-drug formulations, can be a good approach to tackle the problems. Alternatively, drugs can be cleverly designed to inhibit PPIases in specific target cells as a drug–antibody conjugate. Despite all the difficulties, targeting PPIases remains attractive for researchers in the drug discovery field. With the advent of new technology and methods, we believe these ongoing studies are likely to lead to new medications for patients with various disorders in which PPIase plays a role.

## Figures and Tables

**Figure 1 biology-12-01008-f001:**
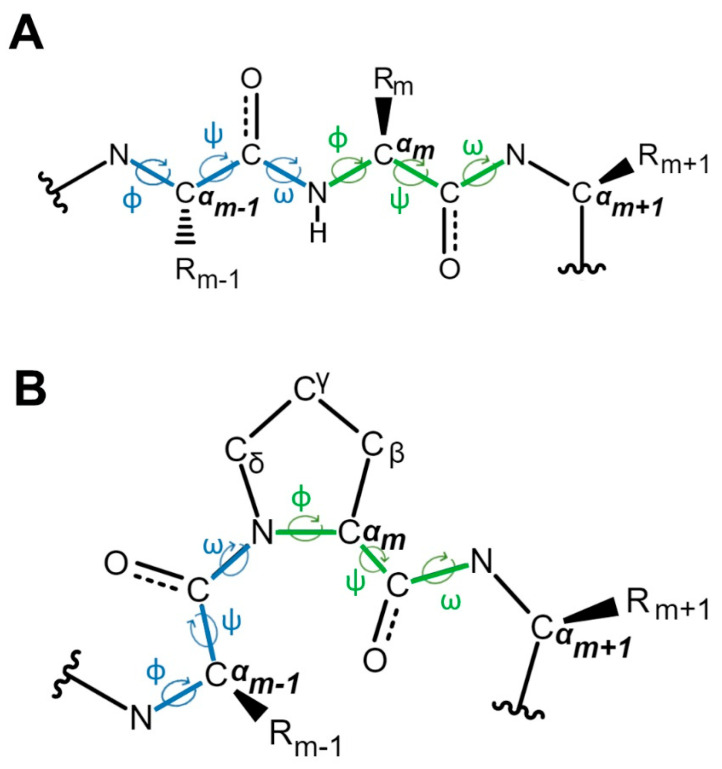
Schematic diagram of protein peptide and the three torsion angles phi (Φ), psi (φ) and omega (ω) that define the conformation of protein backbone. The phi angle is around the -N-C^α^- bond; the psi angle is around the -C^α^-C- bond and the omega angle is around the -C-N- bond, which is also referred to as the peptide bond. Two sets of phi, psi and omega angles are labeled for residue at position m-1 and m in blue and green, respectively. (**A**). Peptide that has linear residues at position m-1, m and m+1. (**B**). Peptide that has a proline at position m. The freedom of the phi angle of proline is restricted due to the ring structure of proline.

**Figure 2 biology-12-01008-f002:**
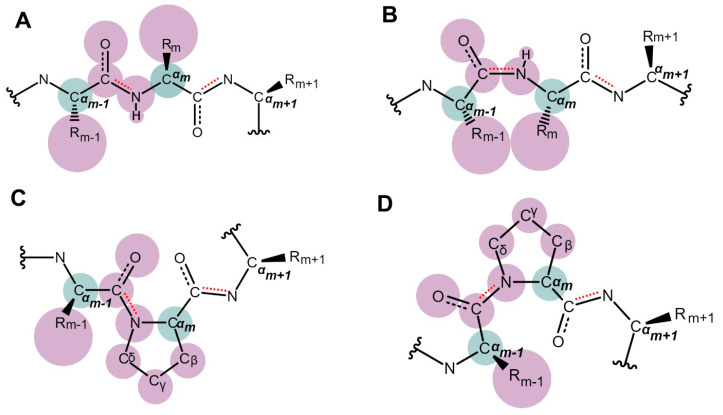
Schematic diagram of *trans*- and *cis*-conformations around the peptidyl bond that connects residues m and m-1. To aid visualization, atoms around this peptidyl bond are depicted by spheres according to their sizes although not strictly proportionally. All spheres are in pink except for the two C^α^ atoms that are painted in green for easy identification. (**A**). *Trans*-conformation around the peptidyl bond connecting two linear residues. In this conformation, C^α^_m-1_ and C^α^_m_ along with their connected linear R groups are dispersed on two sides of the peptidyl bond and therefore this conformation is energetically favorable. (**B**). *Cis*-conformation around the peptidyl bond connecting two linear residues. In this conformation, C^α^_m-1_ and C^α^_m_ along with their connected linear R groups are concentrated on one side of the peptidyl bond, leading to crowdedness, or even clashing between atoms. Therefore, this conformation is energetically unfavorable. (**C**). *Trans*-conformation around the peptidyl bond connecting a linear residue with a proline. In this conformation, although C^α^_m-1_ and C^α^_m_ are located on different sides of the peptidyl bond, due to the cyclic arrangement of atoms on proline, the C^δ^ atom replaces the hydrogen atom and wraps back to the backbone to be close to C^α^_m-1_ and its connected R_m-1_ group. This leads to the atoms being not as dispersed as in scenario (**A**). (**D**). *Cis*-conformation around the peptidyl bond connecting a linear residue with a proline. In this conformation, although C^α^_m-1_ and C^α^_m_ are located on the same side of the peptidyl bond, because the cyclic arrangement of proline loops the side chain atoms back to the backbone, atoms are not as concentrated on one side as seen in scenario (**B**). As a result, the energy difference between the *trans*- and *cis*-conformation around a peptidyl-prolyl bond is much smaller than that of two linear residues.

**Figure 3 biology-12-01008-f003:**
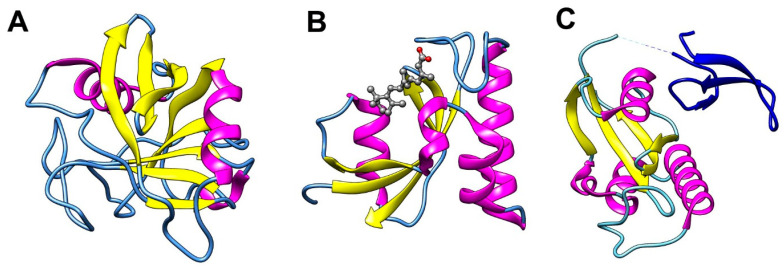
Overall structures of all three types of PPIases. Ribbon representation is colored by secondary structure with yellow, magenta, and blue for helices, sheets, and loops, respectively. (**A**). Overall structure of cyclophilin (PDB code: 1MF8 [53]). (**B**). Overall structure of FKBP12 (PDB code: 1FKJ [54]). The ligand, FK506, is also displayed in ball-and-stick representation. (**C**). Overall structure of Pin1 (PDB code: 2ITK [55]). The PPIase domain is colored by secondary structure while the WW domain is shown by dark blue. The dotted line indicates some missing residues connecting the two domains. All three images are generated by UCSF Chimera [56].

**Figure 4 biology-12-01008-f004:**
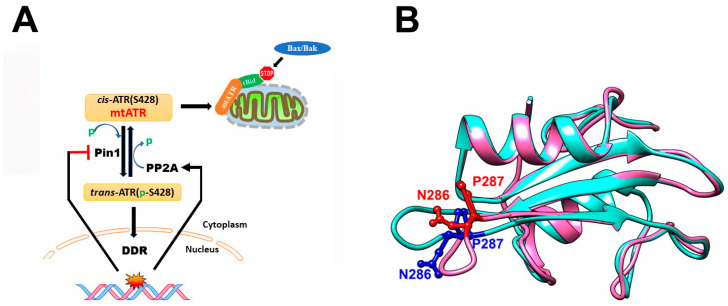
(**A**). Schematic diagram to show how ATR is regulated by proline isomerization to switch between its dual roles in modulating cell death and DNA damage checkpoint signaling. (**B**). NMR structure of *Cis*- and *trans*-proline at position 287 switches the conformation of the Itk SH2 domain. *Cis*- and *trans*-conformations are colored in pink (PDB code: 1LUK [89]) and blue (PDB code: 1LUN [89]), respectively. Proline 287 and its preceding residue Asn286 are shown in ball and stick by red and dark blue in *cis*- and *trans*-conformation, respectively. The image is generated by using UCSF Chimera [56].

**Figure 5 biology-12-01008-f005:**
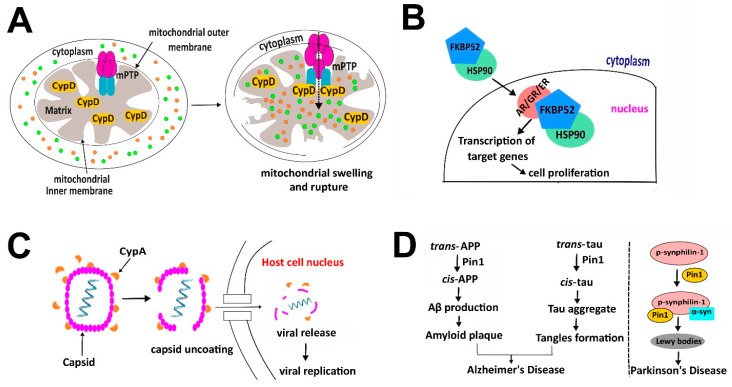
An illustration showing the impact of PPIase in four prominent human diseases. (**A**). Autoimmune disease. In multiple sclerosis, CypD plays a crucial role by binding and regulating mPTP. This interaction leads to the influx of small molecules depicted by green or orange spheres into the cells, causing mitochondrial swelling and rupture, cytochrome C release, and ultimately cell death. (**B**). Cancer. FKBP52 serves as the co-chaperone of HSP90 and interacts with steroid hormone receptors, including the androgen receptor (AR), glucocorticoid receptor (GR), and estrogen receptor (ER). This interaction enhances the transcriptional activity of the respective receptors, leading to increased cell proliferation, as observed in breast and prostate cancer. (**C**). Infectious disease. CypA plays a pivotal role in the life cycle of the HIV-1 virus by catalyzing proline isomerization at the Gly89-Pro90 site of the viral capsid, resulting in its conformational transition that is vital for capsid uncoating, viral genome release, and ultimately promoting viral replication. (**D**). Neurodegenerative disease. Pin1 plays a significant role in the pathogenesis of Alzheimer’s disease and Parkinson’s disease. In the case of Alzheimer’s disease, Pin1 catalyzes the isomerization of APP and tau from *trans* to *cis*, leading to the formation of amyloid plaques and neurofibrillary tangles, respectively, which are distinct hallmarks of Alzheimer’s disease. In the case of Parkinson’s disease, Pin1 interacts with phosphorylated synphilin-1, forming a stable complex with α-synuclein protein and promoting the formation of Lewy bodies, a characteristic feature of Parkinson’s disease. This figure is created using Inkscape (Version 1.2). The RNA icon is created by Servier (Source: Servier, https://smart.servier.com/ (accessed on 1 May 2023)) and is licensed under CC-BY 3.0 Unported (License: CC-BY 3.0 Unported, https://creativecommons.org/licenses/by/3.0/ (accessed on 1 May 2023)).

**Figure 6 biology-12-01008-f006:**
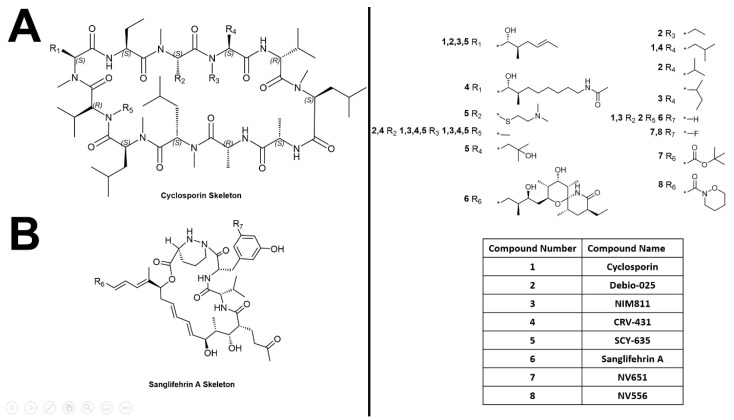
Chemical structures of cyclophilin inhibitors. (**A**). Molecular skeleton of cyclosporin. (**B**) Molecular skeleton of sanglifehrin A. Reference for cyclophilin inhibitors and their corresponding substitutes are shown on the right. Compounds are numbered and shown in bold. Structures of substitutes at different R positions for different compound number are given at the top. Compound names corresponding to compound number are shown in the table at the bottom. The structures are drawn using ChemDraw (Version 22.2.0).

**Figure 7 biology-12-01008-f007:**
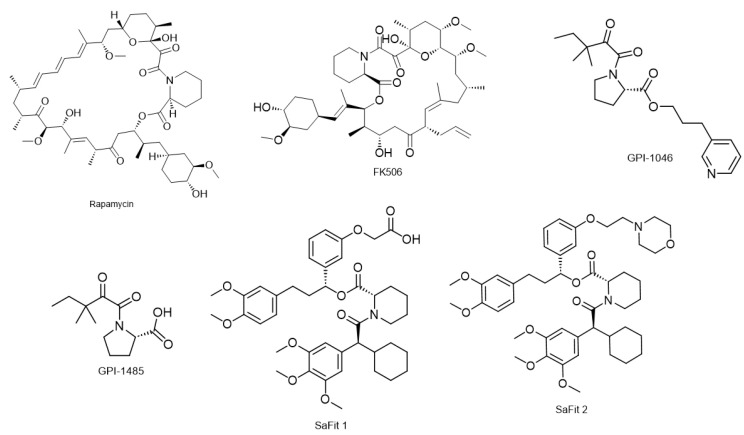
Chemical structures of FKBP inhibitors. Rapamycin and FK506, the two FDA-approved natural ligands for immunosuppression, carry the pipecolate moiety critical for FKBP binding. The two GPI-derivatives carry a similar chemical group to the pipecolate moiety. SaFit1 and SaFit2 are the two inhibitors found to be selective to the FKBP51 isoform. The structures are drawn using ChemDraw (Version 22.2.0).

**Figure 8 biology-12-01008-f008:**
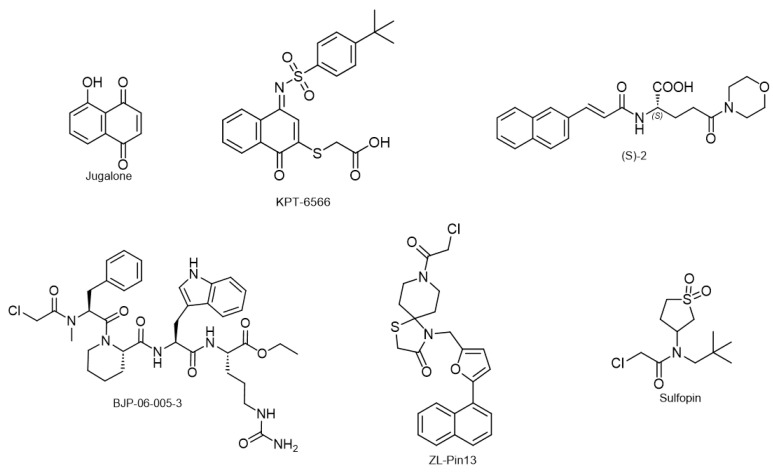
Chemical structures of Pin1 covalent inhibitors. Juglone, the first Pin1 covalent inhibitor, is a natural and simple compound with limited specificity to Pin1. KPT-6566, a derivative of Juglone with added structural complexity, shows higher specificity to Pin1 than Juglone. (S)-2 and BJP-06-005-3 are covalent inhibitors that do not share structural similarity with Juglone. ZL-Pin13 and Sulfopin, also distinct from Juglone, are two newly discovered covalent inhibitors with high selectivity and potency against Pin1. The structures are drawn using ChemDraw (Version 22.2.0).

**Figure 9 biology-12-01008-f009:**
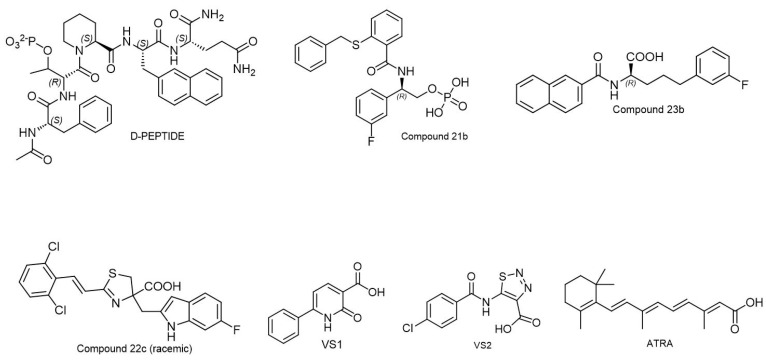
Chemical structures of Pin1 non-covalent inhibitors. D-PEPTIDE is a non-natural peptide. Compound **21b**, Compound **23b**, and Compound **22c** (racemic) developed by Pfizer have diverse structures with high potency inhibiting Pin1 activity in vitro. VS1 and VS2 are two newly discovered inhibitors with smaller molecular sizes. ATRA has been the only Pin1 inhibitor that has moved to clinical trials so far. The structures are drawn using ChemDraw (Version 22.2.0).

**Table 1 biology-12-01008-t001:** Compare and contrast of three families of PPIases.

Prolyl Isomerase	Protein Size	Preferred Proline Site	Triggers	Substrate Proteins	Subcellular Localization
FK506 binding proteins	12–52 kDa	Xaa-Pro	Ligand binding, change in local environment of the protein such as pH	Tau APP, ryanodine receptor, Inositol 1,4,5-triphosphate receptor, TGF-β, heat-shock proteins, nuclear transport receptor including androgen receptor	Cytoplasm (primary), endoplasm reticulum (ER), and nucleus
Cyclophilins	18–45 kDa	Xaa-Pro	Ligand binding, change in local environment of the protein such as pH	HIV1 capsid, NS5A of hepatitis C virus, heat -shock proteins (HSP90), α-synuclein	Cytosol, ER, mitochondria and nucleus
Parvulins	14, 17, and 18 kDa	pSer/pThr-Pro (Pin1) Arg/Leu-Pro (Par14 and Par17)	Phosphorylation on Ser or Thr proceeding to Pro residue (Pin1) Arginine or leucine residues (Par14 and Par17)	ATR, p53, CDC25, α-synuclein, survivin, β-catenin, NF-kβ, Cyclin D1, C-Jun, RNA pol II	Nucleus (Pin1) Nucleus and cytosol (Par14) Mitochondria (Par17)

Note: Xaa = any standard amino acid.

**Table 2 biology-12-01008-t002:** Summary of inhibitors for all three types of PPIases.

Inhibitors	Inhibitor Target	IC50 or Ki	Inhibitor Type	Reference
Cyclosporine	Cyclophilins	420 ± 56 nM	Immunosuppressive	Sedrani et al., 2003 [226]
Debio-025		0.18 ± 0.03 nM	Non-Immunosuppressive Cyclosporine analog	Ptak et al., 2008 [227], Paeshuyse et al., 2006 [228]
NIM811		0.66 μM	Non-Immunosuppressive Cyclosporine analog	Ma et al., 2006 [229]
CRV431		2.5 nM (CypA)	Non-Immunosuppressive Cyclosporine analog	Kuo et al., 2019 [230]
SCY-635		1.84 μM	Non-Immunosuppressive Cyclosporine analog	Hopkins et al., 2010 [231]
Sanglifehrin A		6.9 ± 0.9 nM	Immunosuppressive Sanglifehrin	Sedrani et al., 2003 [226]
NV651		6.3 nM	Non-Immunosuppressive Sanglifehrin analog	Serrano et al., 2021 [232]
NV556			Non-Immunosuppressive Sanglifehrin analog	Kuo et al., 2019 [230], Serrano et al., 2019 [233]
Rapamycin	FKBPs	0.1 nM	Immunosuppressive	Kolos et al., 2018 [61]
FK506		1 nM	Immunosuppressive	Kolos et al., 2018 [61]
GPI-1485			Immunosuppressive	Kolos et al., 2018 [61]
SaFit1		4 nM	Non-Immunosuppressive	Gaali et al., 2015 [234]
SaFit2		8 nM	Non-Immunosuppressive	Gaali et al., 2015 [234]
Juglone	Pin1	7.68 μM	Covalent inhibitor	Zhang et al., 2015 [235]
KPT-6566		0.64 μM	Covalent inhibitor	Campaner et al., 2017 [236]
(S)-2		3.2 μM	Covalent inhibitor	Ieda et al., 2010 [237]
BJP-06-005-3		48 nM	Non-covalent inhibitor	Pinch et al., 2020 [238]
ZL-Pin13		0.067 ± 0.03 μM	Covalent inhibitor	Liu et al., 2022 [239]
Sulfopin		38 nM	Covalent inhibitor	Dubiella et al., 2021 [240]
D-PEPTIDE		20.4 ± 4.3 nM	Non-covalent inhibitor	Zhang et al., 2007 [55]
Compound **21b**		6 nM	Non-covalent inhibitor	Guo et al., 2009 [241]
Compound **23b**		890 nM	Non-covalent inhibitor	Dong et al., 2010 [242]
Compound **22C**		196 nM	Non-covalent inhibitor	Guo et al., 2014 [243]
VS1		>100 μM	Non-covalent inhibitor	Poli et al., 2022 [244]
VS2		19 ± 3 μM	Non-covalent inhibitor	Poli et al., 2022 [244]
ATRA		112 ± 10 μM	Non-covalent inhibitor	Poli et al., 2022 [244]

## Data Availability

Not applicable.

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
