# Peer review of "Proline Isomerization: From the Chemistry and Biology to Therapeutic Opportunities"

_biology, 2023, doi:10.3390/biology12071008_

Round 1
Reviewer 1 Report
The review article on Proline Isomerization by Gurung and coworkers covers an impressive amount of information. The review starts with fundamental biochemical principles and then introduces the reader to some of the most important proline isomerases. It was interesting to read about how many diseases have a connection to proline isomerization. The authors also describe proline isomerase inhibitors. For this section, it would be helpful to have a table organized by inhibitor target, inhibitor name, Ki and inhibitor type (or IC50 if Ki is not available), side effects or other comments and reference. So I suggest to make a table. If this table gets too big, it would still be helpful to show a selection of the most potent inhibitors (or best studied inhibitors).
I also suggest to use a numbering scheme for headings and subheadings (for example 2. Proline Isomerases and 2.1 Cyclophilins). In addition to being consistent with formatting (heading in bold and subheading in bold and italics) this would help to reader to see the organization of the review more clearly.
The pdf document had some minor formatting issues:
1.) Table 1 has a lot of empty space because one of the columns has more text than all the other columns. Maybe re-organize and have a column in which you can write the name of the enzyme, the size, and proline isomerization site underneath each other?
2.) References 66 and 67 have all upper case letters for a title.
3.) The font size was irregular for Figure caption 4 and some subheadings (Perspectives was in a much smaller font than all other subheadings).
no comments
Author Response
This reviewer stated that the review article “covers an impressive amount of information” and “It was interesting to read about how many diseases have a connection to proline isomerization.” We appreciate this reviewer’s positive feedback as well as the informative and detailed revision suggestions to help to improve the manuscript, which has been addressed as listed below.
- This reviewer suggested “it would be helpful to have a table organized by inhibitor target, inhibitor name, Ki and inhibitor type (or IC50 if Ki is not available), side effects or other comments and reference”.
Response: Table 2 on page 47 that summarizes the inhibitors, and their properties has been added as suggested.
- This reviewer suggested to use a numbering scheme for headings and subheadings.
Response: headings and subheadings have been changed accordingly.
- References 66 and 67 have all upper-case letters for a title.
Response: Letter cases for reference 66 and 67 have been changed to be consistent with other references. All other references have also been examined to ensure consistencies.
- The font size was irregular for Figure caption 4 and some subheadings (Perspectives was in a much smaller font than all other subheadings).
Response: Font in Figure 4 caption on page 40 has been changed to be consistent. Font subheadings have been edited to be consistent throughout the manuscript.
Reviewer 2 Report
The authors report a literature review of an interesting topic “Proline Isomerization in therapeutic application”, with several examples.
A review article is a survey of previously published research on a topic. It should give an overview of current thinking on the topic, providing a critical evaluation of the data available from existing studies. In addition, a review provides not only a comprehensive foundation on a topic but highlight key advances that have been made and areas where more focused research may lead to high impact. This point is relevant to include as part of conclusion suggestions for future research on the topic, and focusing what unknowns remains.
The authors collect several examples, but the organization of the review fault to give to readers a present a critical discussion of literature, but only a descriptive summary of different papers on the topics.
Other relevant points:
a) The paper needs an English revision.
b) The nomenclature of the aminoacids atoms need to be revised (example change Ca with Calpha) see IUPAC NOMENCLATURE AND SYMBOLISM FOR AMINOACIDS AND PEPTIDES Pure & Appl. Chem., Vol.56, No.5, p.595—624,1984.
The paper needs an English revision.
Author Response
This reviewer believes that we only “collect several examples, but the organization of the review fault to give to readers a present a critical discussion of literature, but only a descriptive summary of different papers on the topics” along with minor formatting issues.
- This reviewer indicated that we only collected and compiled some examples without being critical. This reviewer indicated that “this point is relevant to include as part of conclusion suggestions for future research on the topic and focusing what unknowns remains”.
Response: With due respect, we disagree with this reviewer’s assessment that we only collected and compiled some examples without being critical. In fact, we made critical evaluations and suggestions on future research and what unknowns remain and urgent studies are needed. The following lists a few examples of such critical evaluations of past studies:
(1). In the formally “Perspective” session on page 21, which has been renamed during revision as “Conclusion” to comply the journal’s requirement, we pointed out that “isoforms and closely related proteins with structural and functional similarities are still poorly understood and require further study. These studies are not only crucial for establishing attractive protein targets for drug discovery and development but will also help to reach higher selectivity and specificity in the inhibitors to achieve the required efficacy and safety profiles for clinical testing.”
(2). In the same section, we discussed that targeting FKBP and Pin1 are not as successful as targeting cyclophilin. We further gave the possible reasons and future directions on how the problems may be solved as quoted below. “The reason could be that targeting FKBP and Pin1 is more complicated due to their essential and heterogenous functions in different tissues such as in neurons and cancer cells, complex network of interacting proteins and pathways being affected, and inhibitors not specific to the isoform.”, and “Tissue-specific drug delivery strategies, such as prodrug or soft-drug formulations, can be a good approach to tackle the problems. Alternatively, drugs can be cleverly designed to inhibit PPIases in specific target cells as a drug-antibody conjugate.”.
(3). In addition, we also integrated different findings and provided our opinions throughout the manuscript. For example, on page 11, we presented overwhelming discoveries that suggest proline isomerization could be the mechanism for 5-HT3 receptor activation. We also presented the opposite opinion that this may not be the case. “However, it has also been suggested that the rates of activation of 5-HT3 receptor is too fast to be driven by the conformational change of Pro303 and isomerization may not be the primary mechanism for channel gating [113].” We then showed that it is possible that the isomerization process is catalyzed by some unidentified PPIases. We then stressed that more studies on this regard are urgently needed.
Nevertheless, to ease this reviewer’s concern on critical evaluation, we revised the manuscript with additional in-depth discussion. For example, on page 14 concerning Pin1’s function in cancer, we added “Despite most studies suggesting Pin1 is pro-cancerous, it has also been observed that Pin1 high expression correlates with a better prognosis in melanoma, prostate, and testis cancer patients, as revealed from a bioinformatics analysis of tumors in the Human Protein Atlas [88]. They suggested that the relationship between p53 and Pin1 expression in cancer cells may explain this phenotype. Pin1-mediated isomerization of wild-type p53 can potentially activate its tumor suppressive functions, whereas mutant p53, which is more prevalent in cancer, may activate its oncogenic function. Taken together, the above findings suggest that Pin1’s function in cancer cells can be complicated and whether Pin1 is pro or against cancer cell growth depends on the tissue and the existence and status of other proteins. Therefore, future research delineating Pin1’s function in different cell types and gene background is greatly needed and would be important to determine Pin1 as a target for cancer therapy, which is also highly relevant to precision medicine.”
- This reviewer suggested to have an English revision.
Response: During revision, the manuscript was professionally edited by Laura Price as acknowledged at the end of the paper on page 21.
- The nomenclature of the amino acids atoms need to be revised (example change Ca with Calpha) see IUPAC NOMENCLATURE AND SYMBOLISM FOR AMINOACIDS AND PEPTIDES Pure & Appl. Chem., Vol.56, No.5, p.595—624,1984.
Response: Format for amino acid atoms has been revised as suggested.
Reviewer 3 Report
The present review is an extensive overview of the research of the last decades focused on the role of proline isomerisation. The authors first summarise historically significant findings regarding the amino acid concentrating on techniques used to study molecular structures, especially cis-trans-conformational changes. Afterwards, PPIs and their respective roles in various cellular functions under physiological and pathophysiologically-altered conditions are explained. Lastly, the authors showcase some PPI inhibitors and their effect on disease progression (e.g., AD) or infection efficiency (e.g., HCV infection). The general outline chosen by the authors is logical, and most figures illustrate the text nicely, adding value to the manuscript.
The manuscript has several formatting errors, including general font size and spacing. Especially the usage of Greek letters (e.g., the legend of Figure 1 or directly on page 1, line 38) and super- and subscripts characters require further attention; the nomenclature of proteins, genes and organisms should also be checked and corrected carefully. The style of references, as well as cursively written words, is not coherent throughout the text. Figure 5 should be revised regarding font size, and the figure legend should explain how to read the table (Sanglifehrin is underlined as an unknown word, please correct that). Abbreviations were either not explained at all, explained more than once throughout the text, or explained after the acronym was already used.
Most importantly, I found that the manuscript sections read as if they were written separately and combined without being proofread once, leading to heavy redundancy. Additionally, most subsections are too crowded with (unnecessary) details that do not add value, making the manuscript excessively long and tiresome to read.
I want to acknowledge the author's effort in the extensive literature research. It is clear that you have invested a significant amount of time and energy into conducting a thorough review of the existing literature on the topic. Therefore, I would recommend a thorough and careful review of the manuscript, removing any repetitive or unnecessary information to improve clarity and overall quality. Additionally, the abstract should be rewritten to be more engaging and avoid repetition of words (significant, various).
In addition to the formatting errors, there are several grammar/spelling errors and sentences that were missing parts or that simply do not make sense. The quality of the language used differs from section to section and should be harmonised.
Author Response
We appreciate very much that this reviewer recognizes the workload of this review article as stated “It is clear that you have invested a significant amount of time and energy into conducting a thorough review of the existing literature on the topic”. This reviewer also provides constructive comments to improve the paper as listed below.
- The manuscript has several formatting errors, including general font size and spacing, the usage of Greek letters and super- and subscripts characters.
Response: Formatting errors including font size and spacing have been corrected throughout the manuscript including figure legends.
- The nomenclature of proteins, genes and organisms should be checked and corrected carefully. The style of references, as well as cursively written words, is not coherent throughout the text.
Response: Nomenclature of proteins, genes and organisms have been carefully checked and corrected. Format of references and italic prints has been examined and corrected to be consistent throughout the manuscript.
- Figure 5 should be revised regarding font size, and the figure legend should explain how to read the table (Sanglifehrin is underlined as an unknown word, please correct that).
Response: Figure 5 has been remade and the figure legend has been rewritten to explain how the table should be used.
- Abbreviations were either not explained at all, explained more than once throughout the text, or explained after the acronym was already used.
Response: Explanation for abbreviations have been added and corrected to ensure consistency.
- Manuscript has redundancy and unnecessary details.
Response: Last paragraph of “Introduction” has repeated information with first paragraph of “Proline Isomerization Catalyzers: Peptidyl-prolyl isomerases” on page 4 and is not necessary. This paragraph has been removed.
- Additionally, the abstract should be rewritten to be more engaging and avoid repetition of words (significant, various).
Response: Abstract has been rewritten to emphasize it is a wholistic view of proline isomerization that aim to fill gaps and bring new awareness of this important translational modification.
- Grammar/spelling errors and sentence and language use not consistent from section to section.
Response: Grammar/spelling errors and sentences and language use have been examined and edited to harmonize from section to section. As discussed above, this manuscript has been professionally edited.
Reviewer 4 Report
In this review, the authors describe the full breadth of topics related to proline isomerization from the basic physical properties of proline and the energetics of isomerization, the role of proline isomerization in protein structure, the connection between proline isomerization and disease, and the potential for therapeutic targeting of proline isomerization. Each of these individual sections builds into the next section to provide a wholistic view of the importance and variable biological roles of proline isomerization. The article is well-written and clearly formatted.
Minor Revisions:
1. Through the first sections of the article, the authors provide clear, well-colored, and thoughtful figures to support their written descriptions, but accompanying figures are completely lacking throughout the subsection on the “Role of proline isomerization in human disease”. Since this section is the most complicated subsection in the article, the article would be improved by the addition of multiple figures within this section to illustrate the role of proline isomerization in this human disease and to provide a visual representation of the complex pathways and regulation presented in this subsection.
2. Figures 5 – 7 need more complete figure legends to accompany these figures with references and descriptions of each of the chemical structures shown in the figures.
3. The layout for Figure 5 with the multiple R-groups clumped over two base structures and with comma separated lists of when R-groups are used in these base structures is confusing and needs revision. Could the authors better group these R-groups with their base structures and help organize these R-groups into logical patterns of substitution and their impact on the cyclophilin activity?
Author Response
We appreciate very much that this reviewer recognizes the uniqueness of this review by pointing out it is “a wholistic view of the importance and variable biological roles of proline isomerization”. The point-by-point response to the constructive comments from this reviewer is listed below.
- The article would be improved by the addition of multiple figures within the section “the role of proline isomerization” in this human disease to provide a visual representation of the complex pathways and regulation presented.
Response: A new figure on page 41 to serve this goal has been added with each panel illustrating the role of proline isomerization for one disease.
- Figures 5 – 7 need more complete figure legends to accompany these figures with references and descriptions of each of the chemical structures shown in the figures.
Response: Figure legends for figure 5-7 (now figure 6-8) have been rewritten.
- Compounds shown in Figure 5 need to be organized with these R-groups in logical manner.
Response: Figure 5, now figure 6, has been remade and figure legend has been re-written to make it easier for the readers to follow
Round 2
Reviewer 2 Report
Aftere the revision the paper can be accepted for publication.